# Targeting of early endosomes by autophagy facilitates EGFR recycling and signalling

Jane Fraser[1], Joanne Simpson[1], Rosa Fontana[1], Chieko Kishi-Itakura[2], Nicholas T Ktistakis[2] & Noor Gammoh[1],* (ID)

## Abstract

Despite recently uncovered connections between autophagy and the endocytic pathway, the role of autophagy in regulating endosomal function remains incompletely understood. Here, we find that the ablation of autophagy-essential players disrupts EGF-induced endocytic trafficking of EGFR. Cells lacking ATG7 or ATG16L1 exhibit increased levels of phosphatidylinositol-3-phosphate (PI(3)P), a key determinant of early endosome maturation. Increased PI(3)P levels are associated with an accumulation of EEA1-positive endosomes where EGFR trafficking is stalled. Aberrant early endosomes are recognised by the autophagy machinery in a TBK1- and Gal8-dependent manner and are delivered to LAMP2-positive lysosomes. Preventing this homeostatic regulation of early endosomes by autophagy reduces EGFR recycling to the plasma membrane and compromises downstream signalling and cell survival. Our findings uncover a novel role for the autophagy machinery in maintaining early endosome function and growth factor sensing.

**Keywords** autophagy; early endosomes; EGFR; galectin; signalling
**Subject Categories** Autophagy & Cell Death; Membranes & Trafficking; Signal Transduction

## Introduction

Vesicular trafficking processes are important in controlling various aspects of cell fate decisions and homeostasis. Of these processes, both endocytosis and autophagy are known to deliver cellular cargoes to lysosomes for degradation and are involved in relaying cues from the extracellular milieu. Recent studies have begun to uncover mechanisms through which the endocytic and autophagic compartments can act in concert to facilitate their optimal activities.

Endocytosis begins with the internalisation of plasma membrane cargoes into early, or "sorting", endosomes marked by specific adaptor proteins and small Rab GTPases [1–3]. These

vesicles can then mature into endosomes destined for various cellular compartments including the plasma membrane (recycling endosomes) or lysosomes (late endosomes), thereby regulating the activities of endocytic cargoes such as receptor tyrosine kinases (RTKs) [4]. Efficient signalling from a subset of endocytosed RTKs has been shown to utilise autophagy-related membranes as signalling platforms [5–7], suggesting that autophagy proteins can regulate pro-growth signalling independent of their degradative function. This is of particular interest given that the perturbed activities of many RTKs, including the epidermal growth factor receptor (EGFR), have been associated with various cancers and developmental defects [8–11].

Autophagy-mediated degradation of various cellular components is critical to maintain cellular homeostasis [12]. Recently, autophagy has been shown to target damaged lysosomal membranes and bacteria-containing vesicles, suggesting that it can mediate the degradation of these vesicles [13–15]. Such endomembrane damage is recognised by the Galectin family of proteins that bind glycans on the luminal face of the vesicles [15,16]. Tank-binding kinase 1 (TBK1) is then required to recruit autophagy players, including WIPI2 and ATG16L1, to the damage site where they facilitate the ubiquitin-like conjugation of ATG8 paralogues, such as LC3, to phosphatidylethanolamine and subsequently autophagosome biogenesis [16,17]. While the role of LC3 in cargo recognition by binding to receptor proteins has been well studied, increasing evidence suggests that upstream autophagy players can also play an important role in recruiting autophagosome substrates [18,19].

Autophagy and the endocytic pathway share multiple commonalities [20]. For example, phosphoinositide-3-phosphate (PI(3)P) is essential for the proper function of both early endosomes and autophagosomes [3,21–23]. Its biogenesis from phosphoinositol is mediated by vacuolar protein sorting-associated protein 34 (VPS34), also known as phosphoinositide-3-kinase class III (PIK3C3) [24]. VPS34 can generate PI(3)P on either endocytic or autophagic membranes by existing in two distinct complexes: the "autophagic" complex I (comprised of ATG14, Beclin-1 and VPS15) and the "endocytic" complex II (comprised of UVRAG, Beclin-1 and VPS15) [25–28]. These complexes can be regulated by the availability of growth factors and nutrients [29–31]. The endocytic pathway can

---

1  Cancer Research UK Edinburgh Centre, Institute of Genetics and Molecular Medicine, University of Edinburgh, Edinburgh, UK
2  Signalling Programme, Babraham Institute, Cambridge, UK
   *Corresponding author. Tel: +44 1316 518526; E-mail: noor.gammoh@igmm.ed.ac.uk

also contribute to autophagosome biogenesis as a number of autophagy players (such as ATG16L1, ATG9, WIPI2 and ULK1) can localise to Rab11-positive (Rab11[+]) recycling endosomes [32–34]. Furthermore, endocytic regulators (e.g. VAMP3, Rab7, CHMP2A and STX17) have been found to be required for autophagosome maturation, closure and their eventual fusion with lysosomes [35–38]. While extensive studies highlight the contribution of the endocytic pathway to autophagosome formation [20], less is known on how autophagy is required for the optimal function of endosomes.

Here, we uncover a novel role for autophagy in maintaining endosomal homeostasis by recognising and targeting perturbed early endosomes. A consequence of inhibiting this process is the disruption of EGFR endocytic sorting which impacts overall cellular response to growth factor signalling.

# Results

## Loss of autophagy perturbs EGFR endocytic trafficking

Given the close interplay between endocytic and autophagic vesicular trafficking, we were interested to understand how the loss of autophagy can impact the endocytic pathway. To test this, we used EGFR as a model endocytic cargo as its regulation has been extensively studied [39]. Glial cells transformed by *Ink4a/Arf* deletion as well as *Tp53* and *Nf-1* knockdown (sh*Nf-1*/sh*Tp53*) were used as a model system [40,41], and CRISPR/Cas9-mediated gene editing of autophagy-related genes (denoted as sg*Atg*) was introduced to inhibit autophagy (Fig EV1). To observe the overall endocytic trafficking of EGFR in real time, Alexa 555-labelled EGF (555-EGF) was monitored in cells by live spinning disc confocal microscopy. Tracking the paths of these vesicles revealed that while EGF exhibited a dynamic trafficking pattern in control cells, it accumulated in the perinuclear region of *Atg7* knockout cells (Fig 1A). Immunofluorescence staining of EGFR confirmed its perinuclear accumulation in the absence of ATG7 after 15 min of EGF treatment (Fig 1B and C). To further investigate this disrupted endocytic trafficking, we assessed changes in ligand–receptor colocalisation [42] and observed an increased percentage of EGFR vesicles that remained positive for 555-EGF in autophagy-deficient cells (Fig 1D and E). This elevated ligand–receptor binding is suggestive of a perturbed trafficking of EGFR at early endosomes. To test this, we assessed the colocalisation between EGFR and the early endosome marker Rab5. Figure 1F and G shows that at early time points (5 min), EGFR occupancy in Rab5[+] endosomes was comparable between control and sg*Atg7* cells, indicating that early endocytic uptake of EGFR from the plasma membrane is not affected by ATG7 loss. However, at a later time point (15 min), EGFR residency in early endosomes was strikingly increased in sg*Atg7* cells compared to control cells. Overall, these data suggest that autophagy inhibition alters EGFR trafficking resulting in its accumulation at early endosomes.

## Autophagy inhibition increases PI(3)P-positive early endosomes

To further investigate a potential defect in early endosomes upon autophagy deficiency, we examined the levels of PI(3)P, a key phosphoinositide determinant of autophagosome biogenesis and early

endosome function [2,43]. To do this, we employed a post-fixation technique using an Alexa 488-conjugated recombinant FYVE-domain probe in order to avoid interference with endosomal trafficking as a result of PI(3)P-binding domain overexpression in cells [44]. Interestingly, inhibition of autophagy in glial cells by ATG7 or ATG16L1 deletion (Fig EV2A) resulted in increased overall cellular levels of PI(3)P (Figs 2A and B, and EV2B). We further tested whether this increase in PI(3)P levels was associated with early endosomes marked by the PI(3)P effector EEA1 [2] and observed a significant increase in PI(3)P[+] EEA1 vesicles in sg*Atg7* and sg*Atg16l1* glial cells (Fig 2A and C). Similar results were obtained in sg*Atg7* mouse embryonic fibroblasts (MEFs), suggesting that changes in early endosomal PI(3)P as a result of autophagy inhibition are consistent in other cell types (Fig EV2C and D). As observed with Rab5 (Fig 1F and G), EGFR exhibited a higher residency in EEA1[+] endosomes upon autophagy inhibition in glial cells (Fig 2D and E).

The production of PI(3)P catalysed by VPS34 is regulated by adaptor proteins that form part of the endocytic or autophagic VPS34 complexes [25–28]. To test whether changes in PI(3)P levels upon autophagy inhibition were due to differences in the binding of VPS34 to its adaptor proteins, immunoprecipitations of endogenous VPS34, or its binding partner Beclin-1, were performed from control or sg*Atg7* cells. The interactions between VPS34 and its complex components, including Beclin-1, Rubicon, UVRAG and ATG14, were unaltered by autophagy status (Figs 2F and G, and EV2E). Similar results were obtained in MEF cells (Fig EV2F). Together, these findings suggest that the compositions of VPS34 complexes are not altered upon the deletion of autophagy genes.

## Early endosomes are disrupted in the absence of autophagy machinery and stain positively for Gal8

To further examine the cause of EGFR accumulation at Rab5[+] and EEA1[+] endosomes, we tested the possibility that damaged early endosomes accumulated in cells that lack autophagy. Recognition of damaged endomembranes can be mediated by a family of glycan-binding proteins known as Galectins that act as "eat-me" signals [45]. Of these, Gal3, Gal8 and Gal9 have been involved in the recognition of damaged lysosomes and salmonella-containing vesicles [13–15]. To assess whether autophagy-deficient cells accumulate damaged early endosomes that may be recognised by these Galectins, we expressed YFP-tagged Gal3, Gal8 or Gal9 in glial cells and their colocalisation with EEA1 was compared between control and sg*Atg7* cells. Figure 3A and B shows that while the colocalisation between EEA1 and Gal3 or Gal9 was not significantly affected by ATG7 depletion, Gal8 localisation to early endosomes was strongly increased upon ATG7 loss. This suggests that damaged early endosomes may accumulate when autophagosome maturation is inhibited by ATG7 deletion. Moreover, Gal8 labelled EGF[+] vesicles in ATG7-deficient cells, indicating that EGFR can localise to this subset of damaged early endosomes (Fig 3C and D). Disruption of early endosomal function can also be induced chemically by monensin treatment in control glial cells resulting in enhanced localisation of Gal8 at early endosomes (Fig 3E and F) [46]. Together, these data suggest that damaged early endosomes can accumulate in the absence of autophagy genes or during chemical disruption of endomembranes.

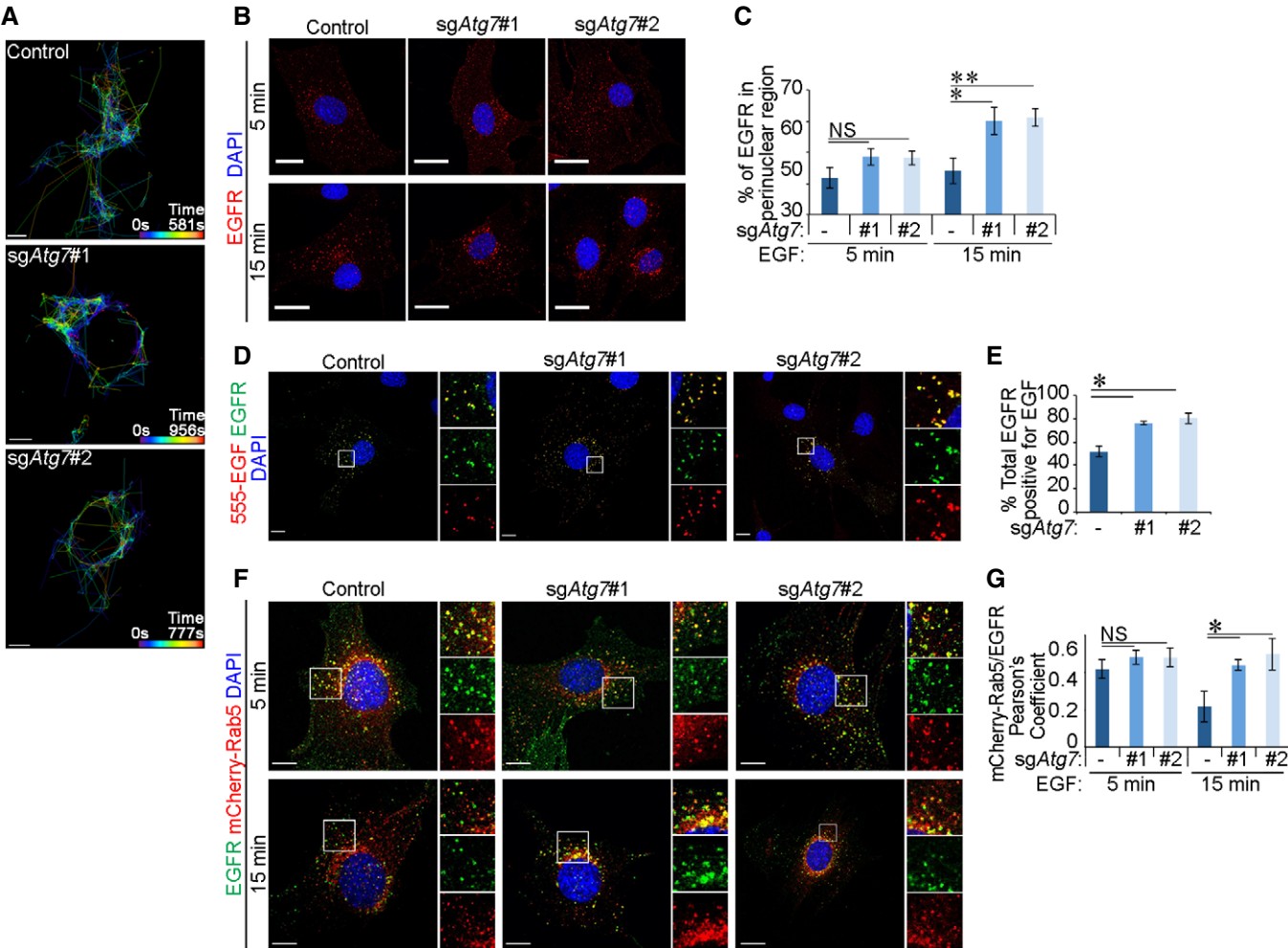

**Figure 1. Loss of autophagy perturbs EGFR endocytic trafficking.**

The following experiments were performed in glial sh*Nf-1*/sh*Tp53* glial cells serum starved for 4 h before assaying. Cells were expressing either Cas9 alone (control) or Cas9 and sgRNA targeting *Atg7* (sg*Atg7* #1 or #2).

A   Spinning disc confocal live cell imaging of Alexa 555-EGF (555-EGF) shown as vesicle tracking with time represented as a colour spectrum. Tracking started 5 min after addition of 20 ng/ml 555-EGF for the indicated durations. Scale bar: 10 μm.
B   Immunofluorescence staining of EGFR following 5- or 15-min stimulation with 20 ng/ml EGF. Scale bar: 20 μm.
C   Quantification of EGFR vesicles in a perinuclear region (within 30 μm diameter of the centre of the nucleus) (in B).
D   Cells were stimulated with 20 ng/ml 555-EGF for 15 or 30 min before immunofluorescence staining against EGFR. Scale bar: 10 μm.
E   Quantification of percentage of total EGFR vesicles that colocalise with 555-EGF (in D).
F   Cells stably expressing mCherry-Rab5 were stimulated with 20 ng/ml EGF for indicated times before immunofluorescence staining against EGFR. Scale bar: 10 μm.
G   Pearson's colocalisation coefficient between mCherry-Rab5 and EGFR (in F).

Data information: Statistical analyses were performed on at least three independent experiments, where error bars represent SEM and *P* values represent a two-tailed Student's *t*-test: NS *P* > 0.05, **P* < 0.05, ***P* < 0.01.

## Autophagy machinery targets early endosomes

We hypothesised that damaged early endosomes may be recognised by the autophagy machinery facilitating their targeting to the lysosome system. We tested this during monensin-induced endosomal damage in autophagy-competent cells. Indeed, monensin treatment resulted in the labelling of EEA1[+] endosomes with GFP-LC3 stably expressed in control glial or MEF cells but not in those lacking ATG7 (Figs 4A and B, and EV3A and B). The localisation of LC3 to EEA1[+] vesicles suggests that the autophagy machinery may be required to recruit lysosomes to early endosomes. Indeed, EEA1[+] vesicles stained positive for the lysosomal marker LAMP2 following monensin-induced endomembrane stress in control but not sg*Atg7* cells (Fig 4C and D). Interestingly, treatment of cells with bafilomycin A1 to inhibit lysosomal acidification led to the enhanced colocalisation between LC3 and EEA1, suggesting that the lysosomal targeting of early endosomes by the autophagy machinery occurs under basal levels (Fig 4A and B). A potential increase in total early endosome number was not significant between autophagy-competent and autophagy-incompetent cells (Fig EV3C) likely due to high

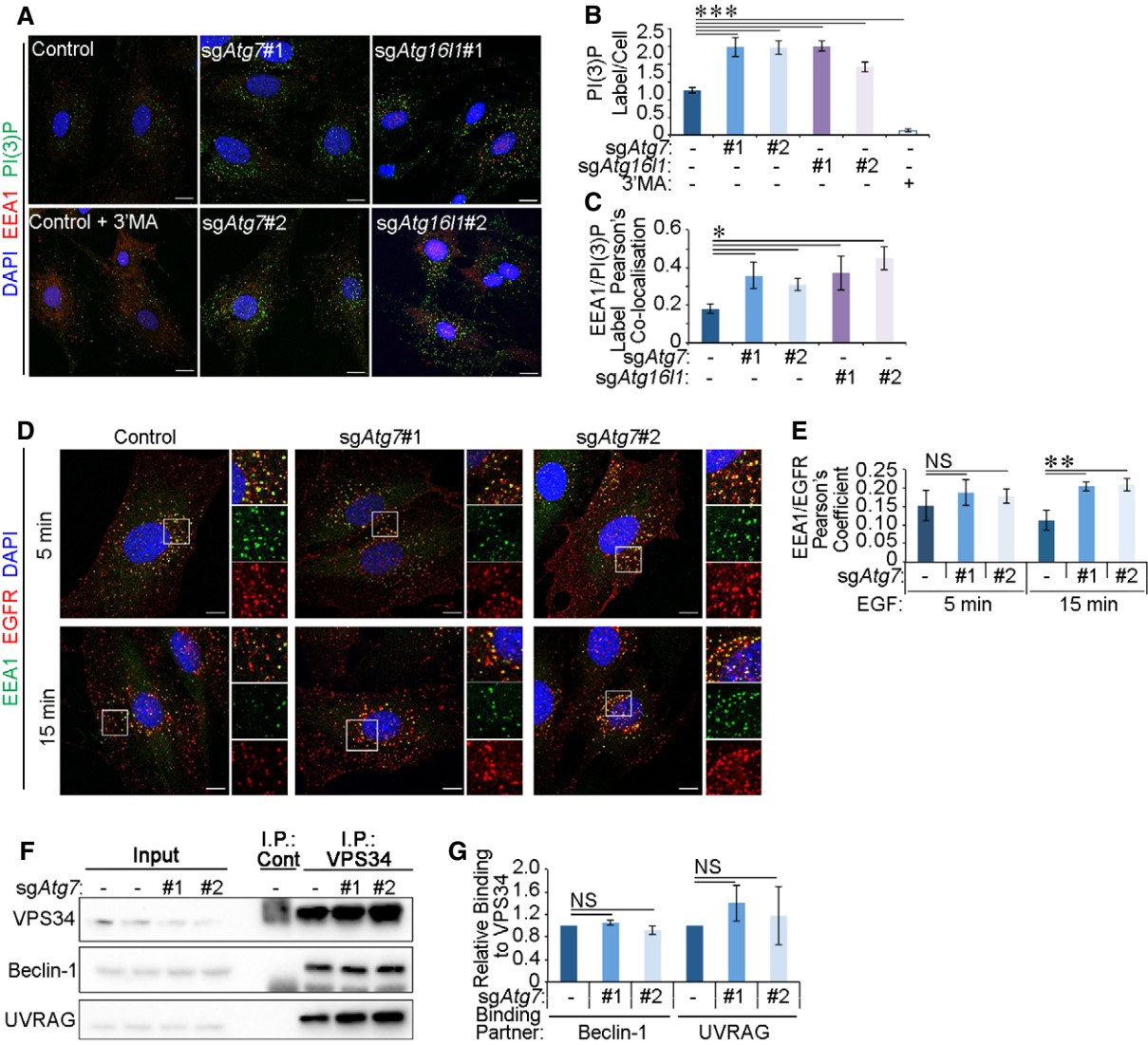

**Figure 2. Autophagy inhibition increases PI(3)P-Positive early endosomes.**

The following experiments were performed in glial shNf-1/shTp53 glial cells serum starved for 4 h before assaying.

A Control, sgAtg7 or sgAtg16l1 cells were treated with 2 ng/ml EGF for 15 min. Cells were then processed for staining using anti-EEA1 antibodies and PI(3)P probe (Alexa 488-labelled 2XFYVE domains). To ensure the specificity of the probe, control cells were pre-treated with 5 mM 3'MA for 30 min. Scale bar: 10 μm.

B Quantification of PI(3)P+ label intensity per cell (in A).

C Pearson's colocalisation coefficient between PI(3)P and EEA1 (in A).

D Control and sgAtg7 cells were stimulated 20 ng/ml EGF for 5 or 15 min before immunofluorescence staining against EEA1 and EGFR. Scale bar: 10 μm.

E Pearson's colocalisation coefficient between EEA1 and EGFR (in D).

F Endogenous VPS34 was immunoprecipitated from control and sgAtg7 cells that were treated with 2 ng/ml EGF for 15 min and then lysed in CHAPS-containing detergent buffer and binding partners detected by Western blotting.

G Densitometry analyses of proteins coimmunoprecipitated with endogenous VPS34 (in F).

Data information: Statistical analyses were performed on at least three independent experiments, where error bars represent SEM and P values represent a two-tailed Student's t-test: NS P > 0.05, *P < 0.05, **P < 0.01, ***P < 0.001.

variations in endosome numbers between cells or the existence of compensatory mechanisms that affect endosome biogenesis or maturation. Altogether, these findings suggest that targeting of early endosomes to lysosomes requires components of the autophagy machinery.

We further interrogated the recruitment of upstream autophagy players to early endosomes. Staining of EEA1 showed no significant

colocalisation with endogenous ATG16L1 in autophagy-competent glial cells (Fig 4E and F) consistent with previously published findings [47]. Similarly, induction of autophagy by amino acid starvation did not enhance the colocalisation of EEA1 and ATG16L1 in control cells (Fig EV3D). Intriguingly, a marked increase in the colocalisation between endogenous ATG16L1 and a subset of EEA1+ early endosomes was observed in ATG7-deficient glial cells

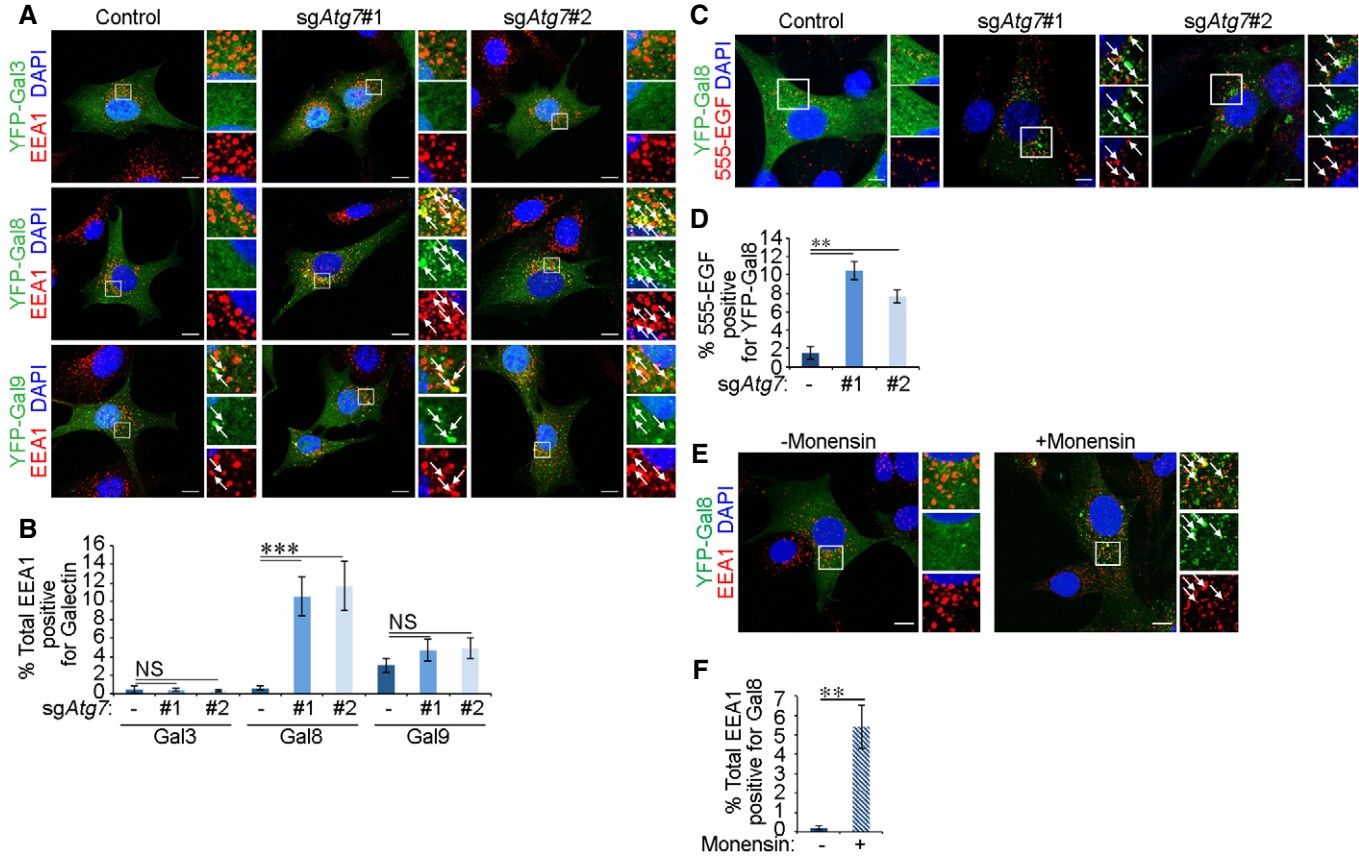

**Figure 3. Early endosomes stain positively for Gal8 upon autophagy inhibition or monensin treatment.**

The following experiments were performed in glial shNf-1/shTp53 glial cells serum starved for 4 h before assaying. Cells were expressing either Cas9 alone (control) or Cas9 and sgRNA targeting Atg7 (sgAtg7 #1 or #2).

A  Cells transiently expressing YFP-Galectins (YFP-Gal3, YFP-Gal8, or YFP-Gal9) were stimulated with 2 ng/ml EGF for 15 min before fixation and immunofluorescence staining against EEA1. Scale bar: 10 μm.
B  Quantification of percentage of total EEA1 vesicles that colocalise with YFP-Galectins (in A).
C  Cells transiently expressing YFP-Gal8 were stimulated with 20 ng/ml Alexa 555-EGF (555-EGF) for 15 min. Scale bar: 10 μm.
D  Quantification of the percentage of YFP-Gal8-labelled vesicles that colocalise with 555-EGF in control or ATG7-deficient cells (in C).
E  Cells transiently expressing YFP-Gal8 were treated 100 μM monensin for 1 h and stimulated with 2 ng/ml EGF for 15 min before fixation and immunofluorescence staining against EEA1. Scale bar: 10 μm.
F  Quantification of percentage of total EEA1 vesicles that colocalise with YFP-Gal8 upon monensin treatment (in E).

Data information: White arrows indicate colocalisation. Statistical analyses were performed on at least three independent experiments, where error bars represent SEM and P values represent a two-tailed Student's t-test: NS P > 0.05, **P < 0.01, ***P < 0.001.

(Fig 4E and F). This was also observed with Flag-S-tagged ATG16L1 (Fig EV3E and F), demonstrating that both endogenous and stably expressed ATG16L1 show comparable localisation [48]. Enhanced colocalisation between ATG16L1 and EEA1 was also observed upon chemically induced endomembrane damage triggered by monensin or Dynasore treatment [46,49,50] (Fig 4G and H). This recruitment is less than that observed in ATG7-deficient cells potentially due to the transient localisation of early autophagy machinery during autophagosome maturation, which is arrested by ATG7 deletion [51]. Importantly, structured illumination microscopy revealed that ATG16L1 localised to a subpopulation of EEA1[+] early endosomes that were also positive for EGF in either sgAtg7- or monensin-treated cells (Fig 4I and J), suggesting that upstream autophagy players can recognise damaged early endosomes where EGFR is arrested.

We explored the possibility that the autophagy machinery may be recruited to early endosomes through an LC3-associated phagocytosis (LAP)-like process which occurs on single membranes requiring the core autophagy machinery (such as ATG7 and ATG16L1) but not PI(3)P, WIPI2 or components of the ULK1 complex [52]. Assessment of endogenous WIPI2 localisation revealed its positive staining on EEA1[+] endosomes in sgAtg7 cells (Fig 4K and L). In addition, a mutant of ATG16L1 that cannot support non-canonical LC3 lipidation on single membranes (ATG16L1[K490A]) [52] was also recruited to EEA1[+] endosomes in glial cells deficient for ATG7 (Fig EV3E and F) or MEFs lacking endogenous ATG16L1 (Fig EV3G and H) further excluding a LAP-like event. Moreover, the association between ATG16L1 and EEA1 was not observed in MEF cells lacking ATG13 (a component of the ULK1 complex required for canonical autophagy but not LAP-like

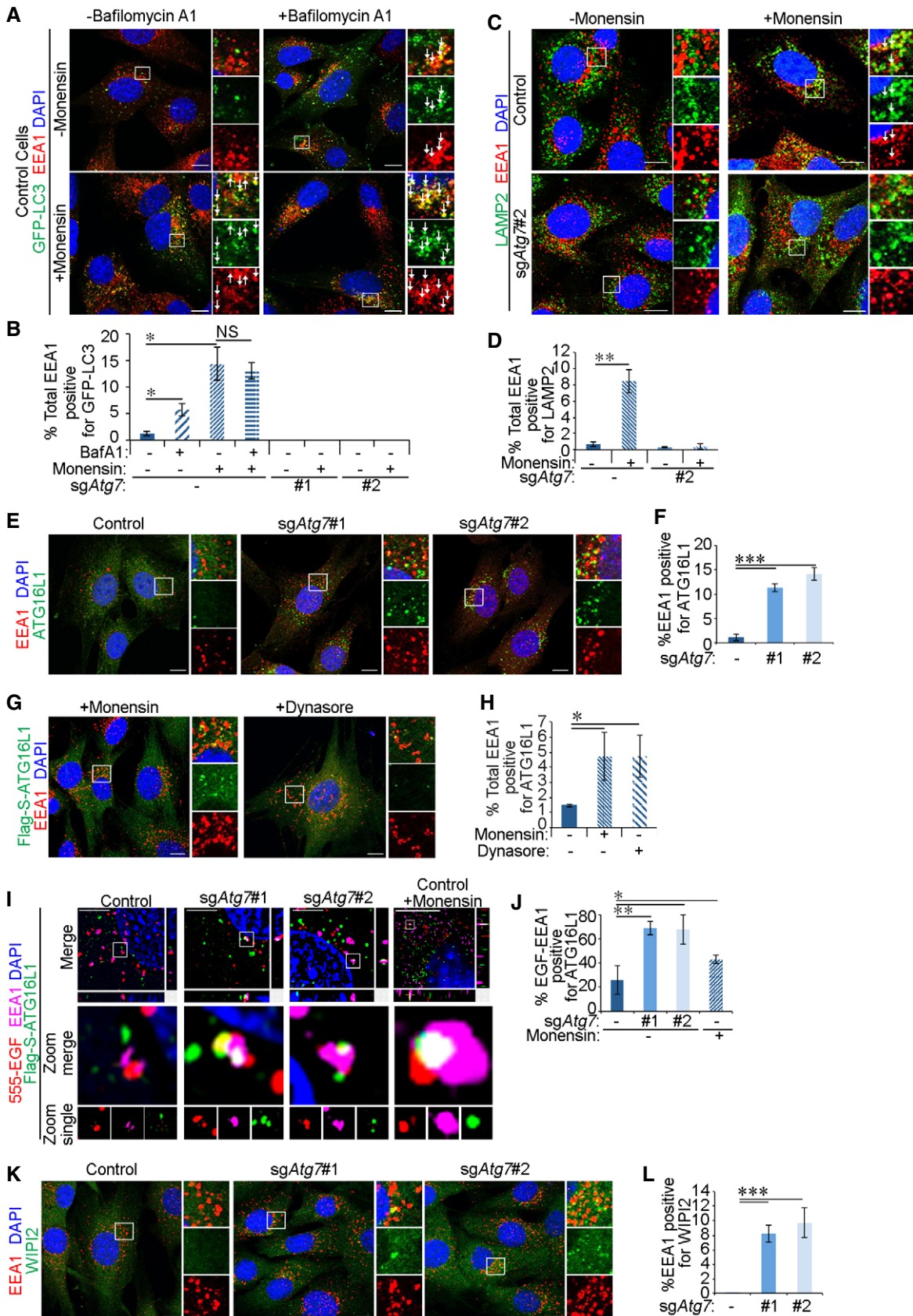

**Figure 4.**

◀

**Figure 4.  Autophagy machinery targets early endosomes.**

The following experiments were performed in glial sh*Nf-1*/sh*Tp53* glial cells serum starved for 4 h before assaying. Cells were expressing either Cas9 alone (control) or Cas9 and sgRNA targeting *Atg7* (sg*Atg7* #1 or #2).

A   Cells stably expressing GFP-LC3 were treated with 100 µM monensin in the presence or absence of bafilomycin A1 (20 nM) for 1 h and stimulated with 2 ng/ml EGF for 15 min before fixation and immunofluorescence staining against EEA1. White arrows indicate colocalisation. Scale bar: 10 µm.

B   Quantification of percentage of total EEA1 vesicles that colocalise with GFP-LC3 (in A).

C   Cells were treated for 1 h with 100 µM monensin and then stimulated with 2 ng/ml EGF for 15 min. LAMP2 and EEA1 were then detected by immunofluorescence staining. White arrows indicate colocalisation. Scale bar: 10 µm.

D   Quantification of the percentage of total EEA1 vesicles that colocalise with LAMP2 (in C).

E   Cells were treated with 2 ng/ml EGF for 15 min before immunofluorescence against endogenous ATG16L1 and EEA1. Scale bar: 10 µm.

F   Quantification of percentage of total EEA1 puncta that colocalise with ATG16L1 (in E).

G   Cells stably expressing Flag-S-ATG16L1 were treated for 1 h with either 100 µM monensin or 30 µM Dynasore, and then stimulated with 2 ng/ml EGF for 15 min. Cells were then stained by immunofluorescence against EEA1 and Flag tag. Scale bar: 10 µm.

H   Quantification of percentage of total EEA1 vesicles that colocalise with Flag-S-ATG16L1 (in G).

I    Untreated control or sg*Atg7* cells, or control cells pre-treated with 100 µM monensin 1 h, stably expressing Flag-S-ATG16L1 were stimulated 20 ng/ml Alexa 555-EGF (555-EGF) for 15 min before fixation and immunofluorescence staining against Flag tag and EEA1. Cells were then imaged by structured illumination microscopy (SIM), and images were reconstructed in Nikon Elements software. Scale bar: 10 µm.

J    Quantification of the percentage of EGF-EEA1 colocalised vesicles that stained triple-positive with ATG16L1 by SIM (in I). Due to the low-throughput nature of this assay, the following cell numbers were counted: control untreated (9 cells), sg*Atg7*#1 (10 cells), sg*Atg7*#2 (9 cells) and control + monensin (11 cells).

K   Cells were treated with 2 ng/ml EGF for 15 min before immunofluorescence against endogenous WIPI2 and EEA1. Scale bar: 10 µm.

L    Quantification of percentage of total EEA1 puncta that colocalise with WIPI2 (in K).

Data information: White arrows indicate colocalisation. Statistical analyses were performed on at least three independent experiments, where error bars represent SEM and *P* values represent a two-tailed Student's *t*-test: *$P < 0.05$, **$P < 0.01$, ***$P < 0.001$.

LC3 lipidation) (Fig EV3I and J) [53]. Finally, the addition of bafilomycin A1, which inhibits the membrane recruitment of LC3 during LAP-like processes [54], did not influence the localisation of GFP-LC3 to early endosomes during monensin treatment (Fig 4A and B). Altogether, these findings uncover a novel process of early endosomal targeting by the canonical autophagy pathway.

## Targeting of early endosomes by the autophagy machinery requires TBK1 activity and Gal8

Previous studies have shown that clearance of damaged endomembranes, as well as selective forms of autophagy, requires the activity of the TBK1 kinase, which phosphorylates autophagic receptors [15,55]. To test whether TBK1 also plays a role during the recognition of damaged early endosomes by autophagy, costaining of active phosphorylated TBK1 (p-TBK1) and EEA1 was performed. EEA1 vesicles stained positively for p-TBK1 in ATG7-deficient cells (Fig 5A and B). Furthermore, treatment with the TBK1 inhibitors MRT68601 [56] and momelotinib [57] ablated EEA1-ATG16L1 colocalisation in sg*Atg7*- and monensin-treated control cells (Fig 5C and D). These findings therefore suggest that TBK1 activity lies upstream of the recruitment of autophagy machinery to damaged early endosomes.

Having observed that damaged early endosomes are recognised by Gal8, we further examined whether the recruitment of the autophagy machinery to early endosomes required Gal8 expression. To do so, we ablated Gal8 expression by CRISPR/Cas9 (Fig 5E) and treated cells with monensin to induce endosome damage. As can be seen in Fig 5F and G, knockout of Gal8 disrupted the colocalisation between ATG16L1 and EEA1 during monensin treatment, suggesting that Gal8 is required for the recognition of damaged early endosomes by the autophagy machinery.

## Autophagy loss disrupts Rab11-mediated recycling of EGFR

We then investigated the consequences of EGFR accumulation in early endosomes during autophagy inhibition. Labelling of cells with fluorescent EGF showed that sg*Atg7* cells exhibited an overall decrease in EGF signal (Figs 6A and B, and EV4A and B), despite the observed increase in EGF and EGFR colocalisation in autophagy-deficient cells compared to their control counterparts (Fig 1D and E). Reduced EGF staining was not due to perturbed lysosomal function [58] (Fig EV4C) or altered EGFR degradation rate (Fig EV4D and E) consistent with the lack of change in EGFR-late endosome colocalisation in the absence of autophagy (Fig EV4F and G). These data indicate that late endosome-to-lysosome trafficking axis is not affected by autophagy loss. Similarly, cell surface protein biotinylation experiments (outlined in Fig EV4H) showed no change in EGFR endocytosis rate in sg*Atg7* or sg*Atg16l1* cells (Fig 6C) in agreement with comparable EGFR-Rab5/EEA1 colocalisation at early time points of EGF stimulation (Figs 1F and 2D). On the other hand, EGF chase experiments revealed that the recycling of EGFR to the plasma membrane was reduced upon the deletion of autophagy genes (Fig 6D). This defective recycling of EGFR reduces receptor availability at the plasma membrane, which, coupled with intact receptor degradation, can lead to diminished intracellular levels of EGF at later time points in autophagy-deficient cells (Fig 6A). Interestingly, EGFR recycling was also reduced in the presence of the endosome damaging agent, monensin, when calculated as a percentage from endocytosed population (Fig 6D). Monensin treatment also perturbed EGFR endocytosis (Fig 6C) likely due to additional effects of this inhibitor. To mimic the recycling defect caused by autophagy loss, EGFRvIII was expressed in glial cells. This oncogenic mutant of EGFR lacks the EGF-binding domain but can heterodimerise with the full-length receptor, thereby altering its endocytic trafficking and increasing downstream signalling [8]. EGFRvIII expression reduced the endocytic trafficking of full-length EGFR (Fig 6C), as previously described [8], and prevented its recycling in control cells to a similar extent as loss of ATG7 or ATG16L1 (Fig 6D).

To characterise the disruption of EGFR recycling in the absence of autophagy, we assessed EGFR colocalisation with the recycling endosome markers Rab4 and Rab11. Although the colocalisation of EGFR with Rab4 was not affected by autophagy inhibition

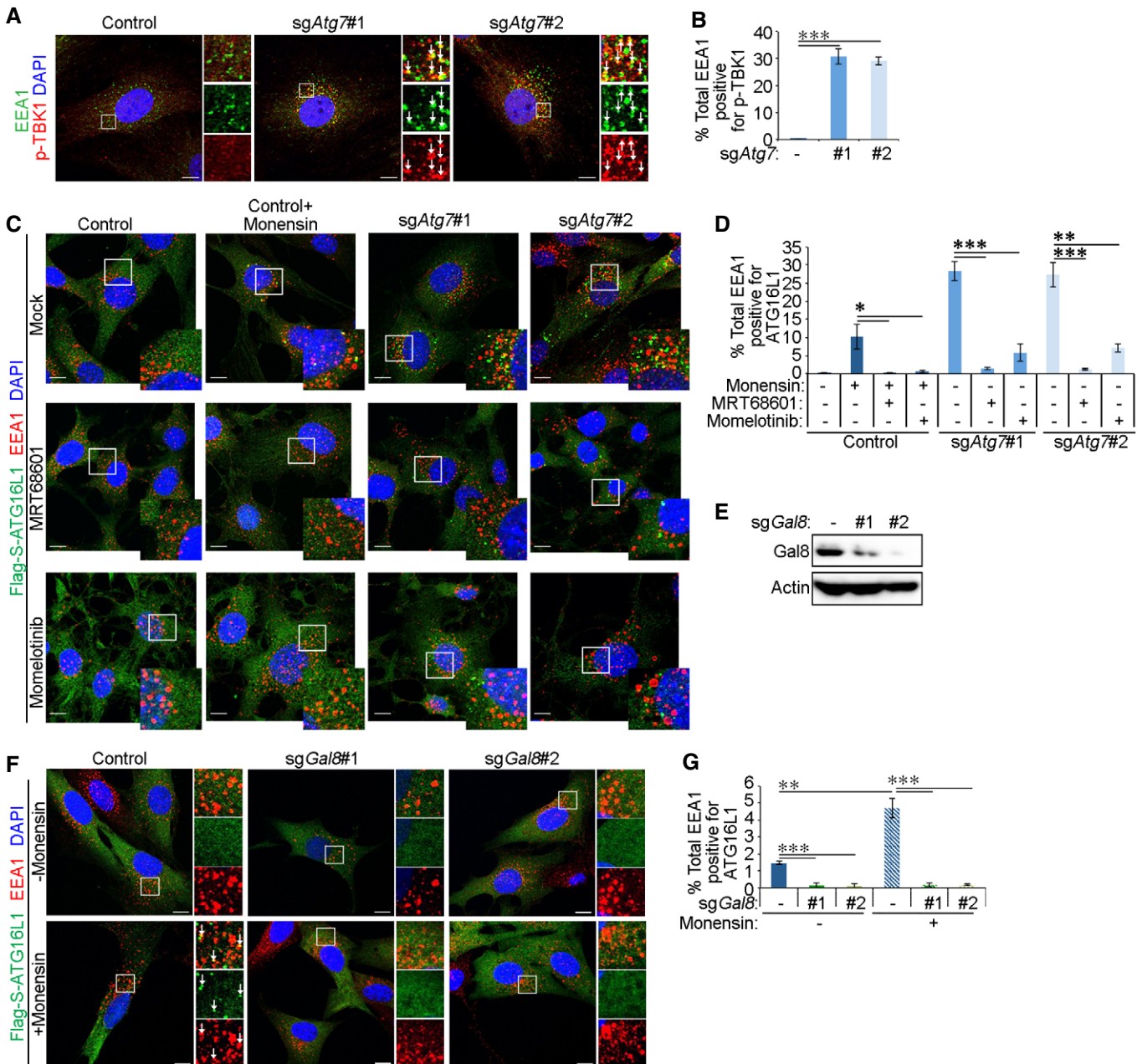

**Figure 5. TBK1 and Gal8 are required for the recruitment of autophagy machinery to early endosomes.**

The following experiments were performed in glial sh*Nf-1*/sh*Tp53* glial cells serum starved for 4 h before assaying. Cells were expressing either Cas9 alone (control) or Cas9 and sgRNA targeting *Atg7* (sg*Atg7* #1 or #2).

A  Cells were treated for 15 min with 2 ng/ml EGF before fixation and immunofluorescence staining against EEA1 and p-TBK1. White arrows indicate colocalisation. Scale bar: 10 μm.

B  Quantification of percentage of total EEA1 vesicles that colocalise with p-TBK1 (in A).

C  Cells stably expressing Flag-S-ATG16L1 were pre-treated for 1 h with TBK1 inhibitors (100 μM MRT68601 or 5 μM momelotinib). Control cells were also treated 100 μM monensin for 1 h as indicated. All cells were stimulated for 15 min with 2 ng/ml EGF followed by fixation and immunofluorescence staining against Flag tag and EEA1. Scale bar: 10 μm.

D  Quantification of total EEA1 vesicles that colocalise with Flag-S-ATG16L1 (in C).

E  Western blotting of sh*Nf-1*/sh*Tp53* glial cells expressing gRNA sequences targeting *Gal8*.

F  Control and sg*Gal8* cells stably expressing Flag-S-ATG16L1 were treated 100 μM monensin for 1 h and stimulated with 2 ng/ml EGF for 15 min before fixation and immunofluorescence staining against EEA1 and Flag tag. White arrows indicate colocalisation. Scale bar: 10 μm.

G  Quantification of percentage of total EEA1 vesicles that colocalise with Flag-S-ATG16L1 in sg*Gal8* cells (in F).

Data information: Statistical analyses were performed on at least three independent experiments, where error bars represent SEM and *P* values represent a two-tailed Student's *t*-test: *$P < 0.05$, **$P < 0.01$, ***$P < 0.001$.

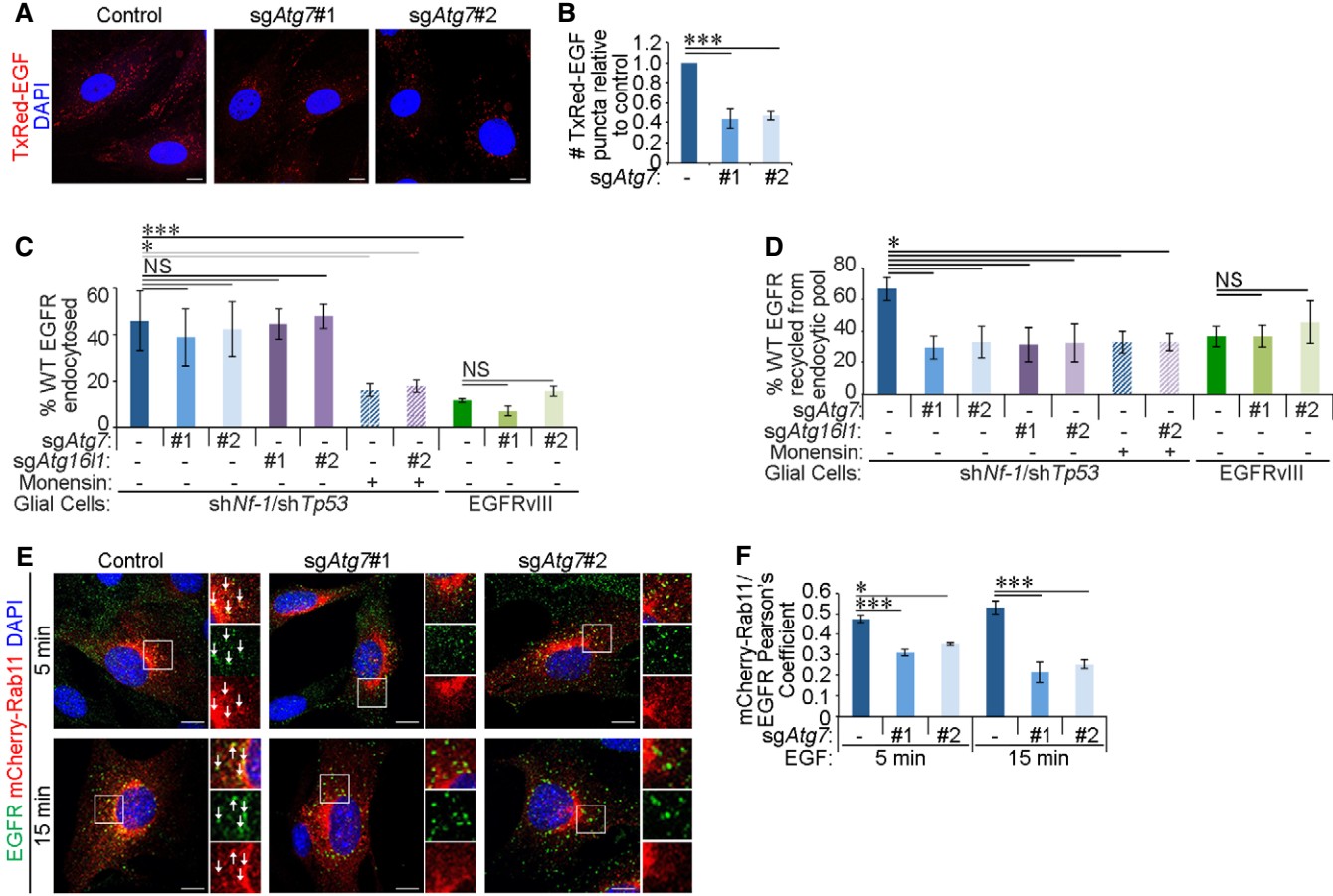

**Figure 6. Autophagy loss disrupts Rab11-mediated recycling of EGFR.**

A  Glial sh*Nf-1*/sh*Tp53* control and sg*Atg7* cells were serum starved for 4 h and then stimulated with 20 ng/ml Texas Red-EGF (TxRed-EGF) for 30 min. Scale bar: 10 μm.
B  Quantification of TxRed-EGF puncta per cell, relative to the number taken up by control cells (in A).
C  Endocytosis rate of wild-type (WT) EGFR was assayed in control, sg*Atg7* or sg*Atg16l1* glial sh*Nf-1*/sh*Tp53* or EGFRvIII-expressing cells by cell surface biotinylation and one application of 2 ng/ml EGF for 15 min. Monensin treatment was added as indicated.
D  Plasma membrane recycling rate of WT EGFR in control, sg*Atg7* or sg*Atg16l1* glial sh*Nf-1*/sh*Tp53* or EGFRvIII-expressing cells was assayed by cell surface biotinylation and successive applications of 2 ng/ml EGF treatments for 15 min (detailed in Fig EV4H). Monensin treatment was added as indicated.
E  Glial sh*Nf-1*/sh*Tp53* control and sg*Atg7* cells stably overexpressing mCherry-Rab11 were serum starved for 4 h, then stimulated with 20 ng/ml EGF for 15 min, and then fixed and stained by immunofluorescence against EGFR. White arrows indicate colocalisation. Scale bar: 10 μm.
F  Quantification of Pearson's colocalisation coefficient between mCherry-Rab11 and EGFR (in E).

Data information: Statistical analyses were performed on at least three independent experiments, where error bars represent SEM and $P$ values represent a two-tailed Student's $t$-test: NS $P > 0.05$, *$P < 0.05$, ***$P < 0.001$.

(Fig EV4I and J), its colocalisation with Rab11 was markedly reduced (Fig 6E and F). This reduced recycling of EGFR in the absence of autophagy was not associated with a general defect in recycling endosomes as assessed by the recycling rate of the transferrin receptor, which was not affected by ATG7 loss (Fig EV4K and L). Together, these data indicate that autophagy is required for the efficient endocytic recycling of EGFR to the plasma membrane via Rab11 recycling endosomes.

## Autophagy is required for the maintenance of EGF-mediated signalling and survival

The recycling of RTKs to the plasma membrane is important for the maintenance of their signalling activities [4]. To test the impact of

reduced EGFR recycling upon autophagy inhibition, we assessed receptor activation and downstream signalling pathways during EGF stimulation in glial cells. Cells lacking the core canonical autophagy players, including ATG7, ATG16L1, ATG3, ATG13 or the cargo recognition protein Gal8, exhibited reduced EGF-induced phosphorylation of EGFR, AKT and ERK (Figs 7A–C and EV5A and B). Importantly, no significant differences in signalling were observed at early time points of EGF stimulation in ATG7-deficient cells, supporting our findings that the initial activation and endocytosis of EGFR were not affected by autophagy loss (Fig 7D and E). Expression of the endocytosis-defective EGFRvIII mutant that enhances downstream signalling [59] restored EGF-mediated AKT phosphorylation in *sgAtg7* cells, thereby suggesting that altering the intracellular trafficking of EGFR can overcome the signalling defect caused by

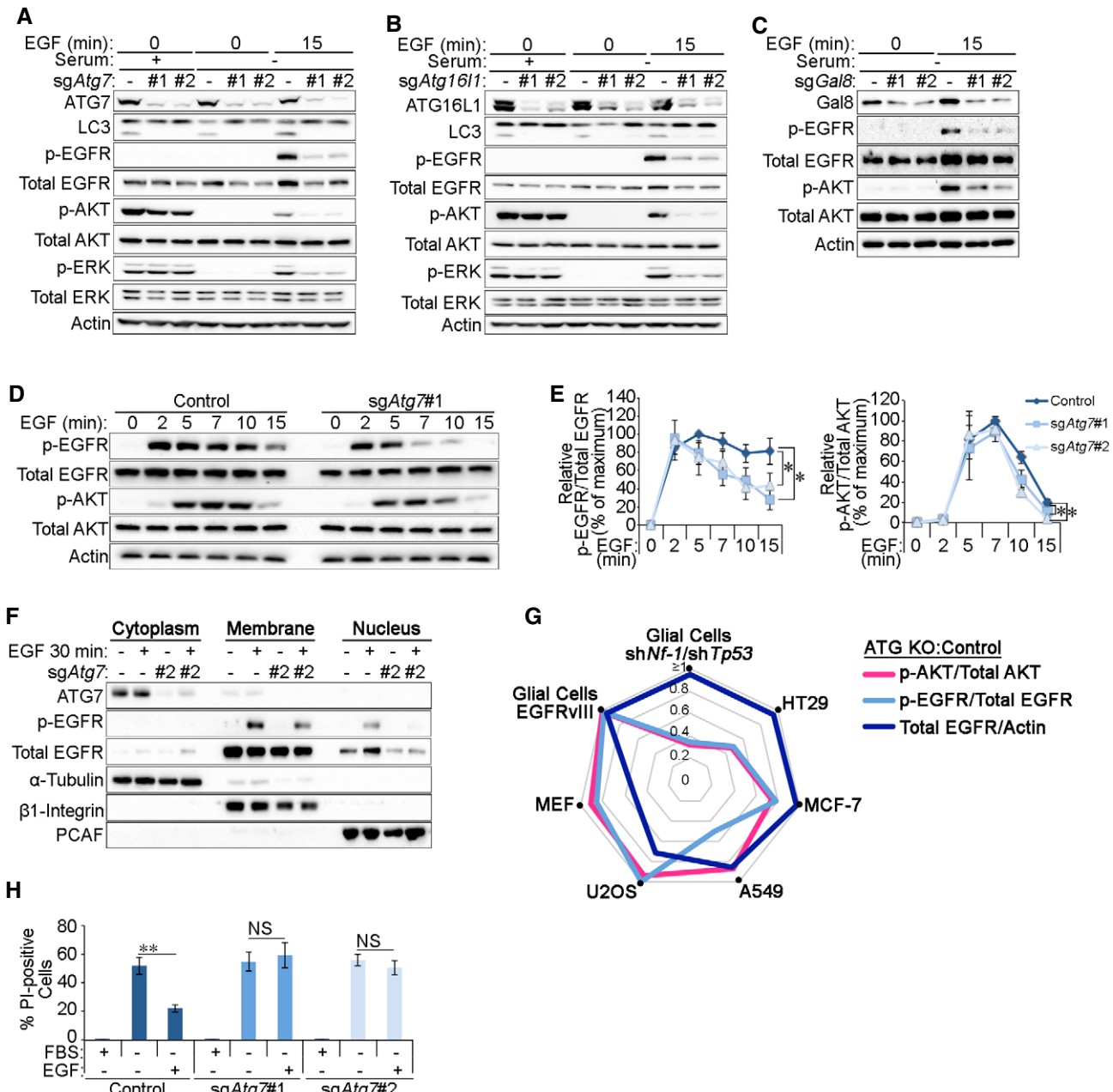

**Figure 7. Autophagy is required for the maintenance of EGF-mediated signalling and survival.**

A–C  Glial shNf-1/shTp53 sgAtg7 (A), sgAtg16l1 (B) or sgGal8 (C) cells were either left untreated (+serum) or serum starved (−serum) for 4 h and then stimulated 2 ng/ml EGF for 15 min before analyses by Western blotting.

D  Glial shNf-1/shTp53 control and sgAtg7 cells were serum starved for 4 h and then stimulated with 2 ng/ml EGF for the indicated times before analysis of EGFR signalling pathway activity by Western blotting.

E  Phosphorylated:total protein ratios for EGFR and AKT calculated from densitometry analyses of Western blots were made relative to their respective points of maximal stimulation (100%) (for EGFR: $t$ = 5 min in control cells and for AKT: $t$ = 7 min in control cells) (in D).

F  Glial shNf-1/shTp53 control and sgAtg7 cells were serum starved for 4 h with or without 20 ng/ml EGF 30-min stimulation before subcellular fractionation and analysis by Western blotting.

G  Radar chart indicating the relative levels of pAKT (pink line), pEGFR (light blue line) or total EGFR levels (dark blue line), in autophagy-deficient cell lines relative to their control counterparts. The following knockouts were introduced: glial cells (sgAtg7), HT29 (crATG7), MCF-7 (sgATG5), A549 (sgATG5), U2OS (sgATG5) and MEFs (Atg16l1−/−).

H  Control or sgAtg7 glial shNf-1/shTp53 cells were serum starved for 24 h in the presence or absence of 20 ng/ml EGF. Quantification of the percentage of propidium iodide (PI)-positive cells is shown.

Data information: Statistical analyses were performed on at least three independent experiments, where error bars represent SEM and $P$ values represent a two-tailed Student's $t$-test: NS $P$ > 0.05, *$P$ < 0.05, **$P$ < 0.01.

ATG7 loss (Fig EV5C). In addition, EGF-induced nuclear translocation of EGFR, which is required for its role as a co-transcriptional activator for pro-growth and survival genes [60], was also reduced by autophagy loss (Fig 7F). Together, these data support the conclusion that the targeting of early endosomes by the autophagy machinery is required for proper cellular responses to EGF stimulation.

Having observed similar early endosome defects in MEFs upon autophagy inhibition, we tested EGF-mediated signalling in these cells. Intriguingly, AKT and ERK phosphorylation during EGF stimulation were comparable in autophagy-deficient $Atg16l1^{-/-}$ MEFs and those reconstituted with wild-type ATG16L1 (ATG16L1$^{WT}$, Fig EV5D). However, EGFR protein levels appeared to rely on functional autophagy in MEFs as total EGFR levels were elevated by the expression of ATG16L1$^{WT}$ in $Atg16l1^{-/-}$ cells but not by an autophagy-deficient mutant that cannot bind ATG5 (ATG16L1$^{\Delta A5}$) [61,62]. Similar reduction in EGFR levels was also observed in autophagy-deficient MEFs lacking ATG7 or ATG3 (Fig EV5E) and are likely due to reduced EGFR plasma membrane localisation that may decrease its stability in the absence of acute EGF stimulation. These observations indicate that the disruption of endosomal homeostasis as a result of autophagy inhibition may impact the EGFR pathway in a cell-type-specific manner. To further investigate this, we tested cellular response to EGF stimulation in a panel of cell lines genetically engineered to inhibit autophagy (Fig EV5F–I). As can be seen in Fig 7G, autophagy inhibition led to a striking perturbation in EGFR phosphorylation or total levels in the majority of lines tested. This suggests that targeting of endosomes by autophagy can affect EGFR trafficking and activation in a panel of cell lines. On the other hand, the downstream Akt activation in response to EGF stimulation differed between the cell lines tested. This variation in signalling response to growth factor stimulation is likely dependent on additional oncogenic mutations that enhance pro-growth signalling.

To test whether reduced EGFR signalling in the absence of autophagy impacted EGF-mediated cell survival, glial cells were starved of growth factors (serum starvation) in the presence or absence of EGF and cell death was measured by propidium iodide staining. Figures 7H and EV5J and K show that while serum starvation-induced cell death can be rescued in control cells by the addition of EGF, survival of ATG7-deficient cells remained unaltered, mirroring their inability to maintain pro-growth signalling. Reduced EGF-mediated cell survival in sg$Atg7$ cells was rescued by the expression of EGFRvIII, thereby indicating that rescuing the signalling defect in autophagy-deficient cells can prevent the onset of cell death (Fig EV5J and K). Taken together, these data demonstrate that quality control of early endosomes by autophagy is important for EGFR trafficking, pro-growth signalling and cell survival.

## Discussion

The results presented here outline a novel role for autophagy in regulating early endosome homeostasis. In the absence of autophagy, damaged or non-functional early endosomes accumulate EGFR and inhibit its recycling back to the plasma membrane. This results in the perturbation of EGF-mediated signalling and survival and underlies a novel mechanism by which autophagy can regulate RTK signalling.

Ligand binding stimulates the activation of RTKs leading to their internalisation via clathrin-dependent or clathrin-independent means [63]. While the former is favoured by low doses of growth factors (such as those used in this study) and primarily results in receptor recycling to the plasma membrane, higher doses largely engage clathrin-independent endocytosis machinery and direct the receptor to the lysosome [4]. After their internalisation, active RTKs occupy pre-early endosomes that are negative for PI(3)P. Production of PI(3)P following the recruitment of the endosome-specific VPS34 complex II represses RTK signalling [3]. Our study suggests that autophagy inhibition can reduce EGF-mediated signalling by prolonging the residence of EGFR on PI(3)P$^+$ early endosomes. Whether the overall increase in PI(3)P levels upon autophagy inhibition have additional consequences in the cell is yet to be determined.

Autophagy inhibition also suppresses EGF-mediated signalling by reducing receptor recycling. Our data demonstrate that EGFR is arrested at early endosomes that fail to mature into Rab11$^+$ recycling endosomes, while trafficking to Rab7$^+$ or Rab4$^+$ endosomes remain intact. Interestingly, autophagy knockdown does not appear to interfere with transferrin receptor recycling thereby suggesting an additional degree of cargo or endosomal population specificity. The mechanisms mediating the maturation of early to recycling endosomes are not fully understood, although it is known that phosphoinositol and Rab switching are required [64,65]. Previous studies comparing the internalisation of EGF and transferrin show that each cargo was incorporated at distinct sites with EGF being associated with vesicles enriched with EEA1 [66,67]. This suggests that autophagy inhibition may affect a subset of early endosomes, where EGFR is localised, that are destined to mature into Rab11$^+$ recycling endosomes. The precise mechanism through which this occurs remains unknown and is suggestive of the existence of unknown regulatory mechanisms governing endosome maturation. As autophagy appears to affect a subset of membrane receptors rather than general recycling, its inhibition may serve as a useful tool to decipher such mechanisms.

Previous studies suggest that ATG16L1 can traffic from the plasma membrane via clathrin-coated pits to recycling endosomes without intermediately localising to early endosomes [33,34,68]. In support of this, our data indicate that the colocalisation between ATG16L1 and early endosomes occurs at low levels and is only augmented upon autophagy inhibition or treatment with endosomal damaging agents. It may be that ATG16L1 traffics through early endosomes en route from the plasma membrane to recycling endosomes but is arrested in early endosomes when damage is recognised. This transient or lack of localisation between ATG16L1 and early endosomes under normal conditions could be important to prevent unwanted degradation of functional early endosomes. How early endosomes can be marked for degradation while recycling endosomes contribute to autophagosome biogenesis is yet to be elucidated.

The specific recruitment of Gal8, but not Gal3 or Gal9, to aberrant early endosomes suggests that these Galectins may differentially recognise a binding partner on the damaged endomembrane. Interestingly, out of these three Galectins, only Gal8 has been shown to bind the mTOR apparatus during lysosome injury [14]. Furthermore, studies have shown that although the carbohydrate recognition domain of Galectins is structurally similar, they differ

considerably in their amino acid compositions potentially leading to the recognition of specific carbohydrate moieties on damaged endosomes by Gal8 but not by other Galectins [69,70]. The precise signal that facilitates the recruitment of Gal8 to damaged early endosomes and the subsequent events that lead to the activation of the autophagy machinery remain to be elucidated.

The recruitment of LAMP2$^+$ lysosomes to perturbed early endosomes in an autophagy-dependent manner is suggestive of an "endosome-phagy" process. Our findings thereby contribute to an expanding picture of organelle homeostasis mediated by autophagy. The selective degradation of ER, mitochondria and ribosomes has been shown to be critical for a variety of cellular processes [71,72]. Targeting of early endosomes by autophagy relies on the recognition of damaged membranes by Gal8- and TBK1-mediated signalling, in a similar mechanism to the clearance of damaged lysosomes or bacteria-containing vesicles [14–16]. We find that the accumulation of damaged early endosomal membranes perturbs EGFR trafficking. Localisation of autophagy proteins to endosomes has also been observed during amphisome formation as well as during direct lipidation of LC3 on single membranes [73, 74]. Whether these phenomena also contribute to the requirement of autophagy proteins in the proper trafficking of EGFR remains to be determined. Interestingly, EGFR knockout animals have neurological phenotypes that resemble autophagy loss [75,76]. Additionally, pro-growth signalling pathways downstream of EGFR, which are known to be key drivers of oncogenesis, have been shown to be reduced in autophagy-deficient tumour models [10,41,77]. Altered RTK trafficking upon autophagy ablation may contribute to these defects. Therefore, it would be interesting to test whether additional plasma membrane endocytic cargoes are also affected by the lack of autophagy. Such mechanisms may provide novel means by which the homeostatic role of autophagy impacts important physiological and pathological processes such as in development, immunity and cancer.

## Materials and Methods

### Cell culture, generation and treatment

The following cell lines were maintained at 37°C and 5% $CO_2$ in Dulbecco's modified Eagle's media (DMEM) supplemented with 10 units/ml penicillin, 100 μg/ml streptomycin, 2 mM L-glutamine and 10% foetal bovine serum (FBS) (all reagents were manufactured by Thermo Fisher Scientific): XFM/*tv-a* glial cells expressing the avian virus receptor, TV-A, and harbouring a deletion in the *Ink4a/Arf* locus [40,41], DF-1 chicken fibroblasts (ATCC, CRL-12203), mouse embryonic fibroblasts (MEFs), human embryonic kidney cells (293T), A549 adenocarcinoma cells (WT and Δ*ATG5*, kind gift from Dr Simon Wilkinson, University of Edinburgh) [78], U2OS osteosarcoma cells, MCF-7 breast cancer cells, *Atg16l1*$^{-/-}$ MEFs (kind gift from Dr Shizuo Akira, Osaka University) [79] and *Atg13*$^{-/-}$ MEFs.

HT29 colorectal cancer cells were maintained at 37°C and 5% $CO_2$ in McCoy's 5A medium supplemented with 10 units/ml penicillin, 100 μg/ml streptomycin, 2 mM L-glutamine and 10% FBS (all Thermo Fisher Scientific).

293T cells were used to generate retroviral particles by transfection with polyethylenimine-based lipopolyplexes. For generation of

RCAS viruses, DF-1 cells were transfected with the respective RCAS vectors using Lipofectamine 2000 (Thermo Fisher Scientific) according to the manufacturer's instructions. Following several passages, DF-1 supernatants containing RCAS viruses were collected and used for 3 rounds of infection in XFM/*tv-a* glial cells. All viruses were filtered and used to infect cells in the presence of 1 μg/ml polybrene (Millipore). Gene expression was tested by Western blotting to confirm enhanced or reduced protein levels. Importantly, pooled glial cells were assayed within 10 passages and were not freeze-thawed.

To genetically inhibit autophagy in the array of cell lines tested in Figs 7 and EV5, several methods were required. *Atg3* and *Atg7* knockout MEFs were generated by CRISPR/Cas9-mediated gene editing using Lipofectamine 2000-mediated transient transfection of Cas9 and gRNA constructs, and pooled cells were confirmed by Western blotting. crRNA/tracRNA complexes targeting *ATG7* sequences (Dharmacon, CM-020112-01-0005) were introduced using DharmaFECT 2 Transfection Reagent (Dharmacon) in Cas9-expressing HT29 cells according to the manufacturer's instructions. For glial cells, MEFs and HT29 cells, CRISPR/Cas9 knockout lines were generated by multiple rounds of transfection/infection (usually up to 3 rounds) in the absence of any antibiotics treatment, FACS sorting or single-cell clone selection, and successful inhibition of autophagy was confirmed by Western blotting. Conversely, *ATG5* knockout MCF-7 and U2OS cell lines stably expressing Cas9 were generated by infection with lentiviral vectors expressing sgRNA targeting *ATG5* sequences followed by puromycin selection (Millipore).

All EGF and transferrin stimulations were preceded by 4-h incubation in serum-free DMEM. EGF was used at 2 ng/ml or 20 ng/ml (as indicated in the figure legends), and fluorescently labelled EGF was used at 20 ng/ml. While lower concentrations of EGF favour receptor recycling and higher concentrations trigger lysosomal degradation [4], the observed effects of autophagy inhibition on the endocytic system were found to be independent of EGF concentrations.

For amino acid starvation, cells were incubated in amino acid-free DMEM for 4 h.

The following inhibitors were used: Dynasore (30 μM, Sigma), monensin (100 μM, Sigma), bafilomycin A1 (20 nM, Cell Signaling Technologies), momelotinib (5 μM, Selleck Chemicals) and MRT68601 (100 μM, Tocris).

### Plasmids

For expression in XFM/*tv-a* glial cells, the following plasmids were used: lentiCas9-Blast (gift from Feng Zhang, Addgene plasmid #52962), RCAS-Y vector (gift from William Pavan, Addgene #11478), MSCV-EGFRvIII (gift from Alonzo Ross, Addgene #20737), YFP-Galectin-3, YFP-Galectin-8 or YFP-Galectin-9 (kind gifts from Felix Randow: Thurston *et al*, 2012), mCherry-Rab4 (gift from Michael Davidson, Addgene #55125), mCherry-Rab5 and mCherry-Rab11 (gift from Michael Davidson, Addgene #55124).

For the generation of the RCAS-shRNA vectors, shRNA sequences were initially cloned into pSUPER.retro vectors (OligoEngine, VEC-PRT-0002) [41]. The shRNA and *H1* promoter sequences were then amplified by PCR and cloned into RCAS-Y vectors using NotI and PacI restriction sites. The shRNA target sequences are as follows: RCAS-sh*Nf-1* (5′ CAAGGAGTGTCTGATCAAC), RCAS-sh*Tp*

53 (5′ GTACATGTGTAATAGCTCC) and RCAS-sh*Atg13* (5′ GAGAA GAATGTCCGAGAAT).

For CRISPR/Cas9-mediated gene editing using RCAS-Y vectors, sgRNA sequences were cloned into sgRNA-expressing vectors (gift from George Church, Addgene #41824) as previously described [80]. The sgRNA and *U6* promoter sequences were then amplified by PCR and cloned into RCAS-Y vectors using NotI restriction site. The following sgRNA targeting sequences were used: sg*Atg3* #1 (5′ ATGTGATCAACACGGTGAA, sg*Atg3* #2 (5′ GTTTACACCGCTTGTA GCA), sg*Atg7* #1 (5′ TCACAGGTCCCCGGATTAG), sg*Atg7* #2 (5′ GA AACTTGTTGAGGAGCAT), sg*Atg16l1* #1 (5′ CGAACTGCACAAGAA GCGT), sg*Atg16l1* #2 (5′ AAAGCATGACATGCCAAAT), sg*Gal8* #1 (5′ AGGCGTTGTCCTGTTGTTT) and sg*Gal8* #2 (5′ GGCTGCCTGCTT CACATAC). RCAS vectors expressing sgRNA sequences were infected into XFM/*tv-a* glial cells expressing lentiCas9. Control cells were generated expressing lentiCas9 alone and mock infected with DF-1 media.

For CRISPR/Cas9-mediated gene editing in MEFs, the sgRNA vectors described above (Addgene #41824) were transfected along a Cas9 vector (hCas9, gift from George Church, Addgene #41815) using Lipofectamine 2000. Control cells were generated expressing Cas9 alone and empty gRNA vector.

For CRISPR/Cas9-mediated gene editing in MCF-7 and U2OS cells, lentiGuide-puro lentivirus was used to target *ATG5* (5′ GAGA TATGGTTTGAATATGA).

Plasmids expressing retroviral pBabe-Flag-S-ATG16L1$^{WT}$, pBabe-Flag-S-ATG16L1$^{\Delta A5}$ (deletion of Atg5-binding domain, residues 1-39) and pBabe-GFP-LC3 were described previously [61]. pBabe-Flag-S-ATG16L1$^{K490A}$ mutant was a kind gift from Oliver Florey (University of Cambridge) [52].

## Cell lysis, Western blotting and immunoprecipitation

For cell lysate analyses, cells were washed twice in ice-cold PBS before lysis in RIPA buffer (10 mM Tris pH 7.5, 100 mM NaCl, 1 mM EDTA, 1 mM EGTA, 0.1% SDS, 1% Triton X-100, 1 mM β-ME, 0.5% sodium deoxycholate and 10% glycerol) supplemented with fresh protease inhibitor cocktail V and phosphatase inhibitors (Fisher Scientific, UK). Lysates were cleared by spinning at 21,000 $g$ for 10 min at 4°C. SDS loading buffer (200 mM Tris pH 6.8, 8% SDS, 40% glycerol, 4% β-mercaptoethanol) was diluted 1:4 into lysates. Following 5-min incubation at 95°C, lysates were separated by SDS–PAGE and then transferred onto nitrocellulose membranes (Bio-Rad). Membranes were blocked with 5% non-fat milk TBST before probing with primary antibodies at room temperature for 2–4 h or at 4°C overnight. Following washes in TBST, membranes were then incubated with HRP-conjugated secondary antibodies for 1 h at RT in milk/TBST and developed under UV light using Clarity™ Western ECL substrate (Bio-Rad, 1705061). Western blotting densitometry analysis was performed in ImageLab (Bio-Rad). The radar chart of pAKT, pEGFR and total EGFR ratios comparing ATG knockout to control cells (Fig 7G) was generated from densitometry quantifications of 3 independent experiments for each cell line (representative blots in Fig EV5).

For immunoprecipitations, cells were lysed in buffer containing either 0.8% CHAPS or 0.5% NP-40 and 150 mM NaCl, 25 mM HEPES pH 7.5, 1 mM β-mercaptoethanol, 0.2 mM CaCl$_2$ and

0.5 mM MgCl$_2$ supplemented with fresh phosphatase and protease inhibitors and the proteasome inhibitor MG132 (50 μM, Sigma).

## Cell fractionation

To obtain cytosolic, membrane and nuclear fractions, cells were serum starved for 4 h and then stimulated with 20 ng/ml EGF for 30 min. Cells were then scrapped in an isotonic buffer (150 mM NaCl, 25 mM HEPES pH 7.5, 1 mM β-mercaptoethanol, 0.2 mM CaCl$_2$ and 0.5 mM MgCl$_2$) supplemented with fresh phosphatase and protease inhibitors. For cytosolic fraction isolation, 1 μg/ml digitonin (Sigma) was added and lysates incubated for 30 min at 4°C before spinning for 1 min at 15,000 $g$. The resulting pellet was washed three times with the isotonic buffer and then resuspended in isotonic buffer containing 0.5% NP-40 (Source BioScience). Lysates were then vortexed and spun at 15,000 $g$ for 1 min. The resulting pellet containing the nuclear fraction was similarly washed and resuspended in SDS loading buffer diluted 1:4 in RIPA buffer followed by mechanical sheering using a syringe. The various cell fractions were then analysed by SDS–PAGE and Western blotting.

## Antibodies

Antibodies against the following targets were used for either Western blotting (WB) or immunofluorescence (IF) as indicated: β-actin (WB: Sigma, A5316), AKT (WB: CST, 9272), pSer473-AKT (WB: CST, 4060), ATG3 (WB: MBL, M133-3), ATG7 (WB: Sigma, A2856), ATG13 (WB: Sigma, SAR4200100), ATG5 (WB: Sigma, A0731), ATG14 (WB: MBL, PD026), ATG16L1 (WB and IF: MBL, PD040), ATG16L1 (WB: MBL, M150-3), Beclin-1 (WB: MBL, PD017), EEA1 (IF: CST, 3288), EEA1 (IF: BD Biosciences, 410456), EGFR (WB and IF: Santa Cruz, sc-03), EGFR (WB: Millipore, 04-290), pTyr1068-EGFR (WB: CST, 2234), ERK (WB: CST, 9102), pThr202/pTyr204-ERK (WB: CST, 4370), FIP200 (IF: Abcam, ab176816), Galectin-3 (WB: Santa Cruz, sc-32790), Galectin-8 (WB: Abcam, ab109519), β1-Integrin (WB: CST, 4706), LAMP2 (IF: Abcam, ab25631), LC3B (WB: Sigma, L7543), NF-1 (WB: Bethyl Labs, A300-140A), PCAF (WB: Santa Cruz, E-08), VPS34/PIK3C3 (WB: CST, 4263), Rab5 (WB and IF: CST, 3547), Rab7 (IF: CST, 9367), Rubicon (WB: MBL, PD027), pSer172-TBK1 (IF: CST, 5483), α-Tubulin (WB: CST, 2144), UVRAG (WB: MBL, M160-3B), WIPI2 (IF: Biorad, 2A2), Flag tag (WB and IF: Sigma, F1804), S tag (WB and IF: Bethyl Labs, A190-135A), anti-rabbit-HRP secondary (WB: CST, 7074), anti-mouse-HRP secondary (WB: CST, 7076), anti-rabbit-Alexa 488 (IF: Invitrogen, A11008), anti-rabbit-Alexa 594 (IF: Invitrogen, A11012), anti-mouse-Alexa 488 (IF: Invitrogen, A1101), anti-mouse-Alexa 594 (IF: Invitrogen, A11032) and anti-mouse-Alexa 647 (IF: Invitrogen, A28181).

## Fluorescence imaging

For immunofluorescence analyses, cells were grown on glass coverslips in a 6-well plate. Twenty-four hours later, cells were treated as indicated. Coverslips were then fixed with 3.7% paraformaldehyde (PFA) in 20 mM HEPES pH 7.5 for 10 min on ice and then 20 min at RT. Cells were permeabilised in 0.1% Triton X-100 in PBS for 5 min at RT. Slides were then incubated in primary antibodies in blocking buffer (1% BSA in PBS) at 37°C for 2–3 h or overnight at 4°C, followed by incubation with Alexa-conjugated secondary

antibodies (Invitrogen) for 1 h at room temperature. Finally, nuclei were stained with 1 µg/ml DAPI (Sigma) for 5 min at RT; then, coverslips were mounted on microscope slides with Prolong Gold Anti-fade (Invitrogen). Images were acquired using either the Leica SP5 confocal microscope or, for super-resolution structured illumination microscopy (SIM), the Nikon N-SIM. SIM images were reconstructed using the NIS Elements software.

Perinuclear EGFR localisation was assessed by staining endogenous EGFR following stimulation with 2 ng/ml EGF. Analyses were performed using ImageJ software by drawing circles around the nucleus with a diameter of 30 µm, which was classed as "perinuclear". The number of EGFR puncta inside or outside this region was measured using the "Analyze particles" feature and a constant threshold for each experiment.

For live cell imaging, cells were plated on glass-bottom plates (World Precision Instruments, FluoroDish FD35-100) and starved of serum for 4 h before the addition of Alexa 555-EGF (555-EGF, Invitrogen E35350). Live cell imaging was performed using the Andor Dragonfly spinning disc confocal microscope. Vesicle tracking was performed by Imaris Tracking software.

To assay EGF uptake, cells plated on either coverslips or in a 12-well plastic-bottomed plate were stimulated with 20 ng/ml of TxRed-EGF and 555-EGF, respectively. Cells were then fixed with 3.7% PFA and stained for DAPI. 555-EGF images were acquired using the ImageXpress high-content plate reader, and TxRed-EGF images were taken by confocal microscopy. To quantify these images, a constant threshold was applied and the ImageJ "Analyze particles" feature was used.

To assess the impact of autophagy loss on lysosomal acidification, LysoSensor (Invitrogen, L7535) was added to cells at 1 µM for 1 h before imaging unfixed cells using confocal microscopy and a 20× objective. Total fluorescence levels were then analysed in ImageJ and normalised to cell numbers followed by applying "Analyze particles" feature.

Transferrin recycling was assayed by stimulating cells with 20 ng/ml Alexa 555-Transferrin (555-Tfn, Invitrogen, T35352) and then chasing with Tfn-free, serum-free media to allow recycling of the Tfn before fixing with 3.7% PFA and staining for Rab11.

For EGF/EGFR colocalisation, serum-starved cells were treated for 4 h before stimulating with 20 ng/ml 555-EGF. Cells were then stained for EGFR, and numbers of colocalised puncta were quantified manually in ImageJ. Total puncta numbers were determined using a constant threshold on the "Analyze particles" feature.

The ImageJ "Coloc2" plugin was used to quantify Pearson's colocalisation coefficients between: EGFR and EEA1, Rab4, Rab5, Rab7 or Rab11; EEA1 and PI(3)P; and 555-Tfn and Rab11. Alternatively, the Imaris Coloc software was used to calculate the Pearson's coefficient between 555-EGF and EEA1, Rab5 or Rab7.

For colocalisation between EEA1 and ATG16L1, WIPI2, FIP200, Galectins, p-TBK1, GFP-LC3 or LAMP2 quantifications of colocalised and total puncta numbers were conducted manually and using the ImageJ "Analyze particles" feature with constant threshold, respectively.

## Post-fixation PI(3)P labelling

PI(3)P staining in cells was performed using a FYVE-Alexa 488 probe (kind gift from Ian Ganley, University of Dundee) as previously described [44]. Briefly, cells plated on coverslips were stimulated with 2 ng/ml EGF stimulation for 15 min. Treatment with 10 µg/ml 3-methyladenine (3'MA, Sigma) for 1 h was used as a negative control. Cells were washed with PBS and placed in glutamate buffer (25 mM HEPES pH 7.4, 25 mM KCl, 2.5 mM MgAc, 5 mM EGTA, 150 mM potassium glutamate). Coverslips were then briefly submerged in liquid nitrogen before thawing at RT and then placed in glutamate buffer. Following washing with glutamate buffer, cells were fixed with 3.7% PFA for 30 min followed by quenching and incubation twice in DMEM with 10 mM HEPES pH 7.4 for 10 min. After blocking in 1% BSA in PBS, cells were processed for immunostaining with the inclusion of FYVE-Alexa 488 (1:300) with the secondary antibody incubation.

## Biotinylation endocytosis/recycling assay

Measurement of EGFR endocytosis and recycling was adapted from previously described protocols (outlined in Fig EV4H [81,82]). Briefly, serum-starved cells (4 h) were washed twice with ice-cold PBS and labelled with 0.2 mg/ml biotin (EZ-Link sulfo-NHS-SS-biotin, Thermo Fisher) for 30 min at 4°C. Excess sulfo-NHS-SS-biotin was washed off with PBS and then quenched with 20 mM glycine for 15 min at 4°C. EGFR endocytosis was induced by treatment with 2 ng/ml EGF for 15 min at 37°C and then stopped with ice-cold PBS washes. To strip biotin remaining on the cell surface, a reduction reaction was performed using a buffer containing 37.5 mM NaOH, 37.5 mM NaCl, 25 mM L-glutathione, 25 mM MesNA and 0.5% BSA for 30 min at 4°C, which was quenched with 20 mM iodoacetamide (Sigma) in PBS 20 min at 4°C. Receptor recycling was then performed by incubating cells at 37°C in the presence of EGF for a further 15 min. The stripping procedure was then repeated. To isolate biotinylated proteins, cells were lysed in RIPA buffer before or after the recycling step, and then incubated overnight at 4°C with Streptavidin-sepharose high-performance beads (GE Healthcare 17-5113-01). Beads were washed three times with RIPA and then once with RIPA supplemented with 300 mM NaCl final concentration, followed by two washes with PBS. Beads were then analysed by SDS–PAGE and Western blotting.

## EGF rescue of cell death assay

To investigate growth factor starvation-induced cell death, cells were plated in a 12-well dish and serum starved in the presence or absence of 20 ng/ml EGF. Twenty-four hours later, propidium iodide (1 µg/ml, #10008351, Cayman) was added to cells and analysed using a Tecan Spark20M plate reader. Cells were then fixed with 3.7% PFA and total cell number was obtained by staining with DAPI and measurement using the Tecan plate reader. The ratio of propidium iodide/DAPI relative fluorescence was then calculated to obtain relative values of cell death. Representative cell images were taken on a Leica DM IL LED microscope using Qimaging Retiga EXi Fast1394.

## Graphs and statistical analyses

All chart- and graph-making and statistical analyses were performed using Microsoft Excel. A minimum of three independent experiments were used in each analysis, and statistical significance was

measured using an unpaired 2-sided Student's *t*-test with significance set at: NS $P > 0.05$, *$P < 0.05$, **$P < 0.01$, ***$P < 0.001$.

Expanded View for this article is available online.

## Acknowledgements

We would like to thank Ian Ganley (University of Dundee), Oliver Florey (University of Cambridge) and Felix Randow (University of Cambridge) for providing reagents. We thank Simon Wilkinson for critical reading of the manuscript. We are grateful to members of the N.G. laboratory for discussions and critical reading of the manuscript. N.G. is supported by a Cancer Research UK fellowship (C52370/A21586).

## Author contributions

JF, JS, RF & CK-I performed the experiments; NTK provided critical input; JF and NG designed the study, analysed the data and wrote the manuscript.

## Conflict of interest

The authors declare that they have no conflict of interest.

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
