## [Review Process File · EMBO Reports]

Targeting of Early Endosomes by Autophagy Facilitates EGFR Recycling and Signalling

Jane Fraser, Joanne Simpson, Rosa Fontana, Chieko Kishi-Itakura, Nicholas T Ktistakis, Noor Gammoh

Review timeline:

Submission date:	16 January 2019
Editorial Decision:	25 February 2019
Revision received:	20 May 2019
Editorial Decision:	5 July 2019
Revision received:	19 July 2019
Accepted:	25 July 2019

Transaction Report:

1st Editorial Decision

25 February 2019

Thank you for the submission of your research manuscript to our journal. We have now received the full set of referee reports that is copied below.

As you will see, the referees consider the findings potentially interesting but they also raise a number of concerns. As the study stands, the link between autophagy and EGFR signaling/recycling remains unclear. The referees are concerned that the physiological role of endophagy has not been shown and neither has direct evidence for endophagy been provided. Moreover, the effect on EGFR recycling has not been sufficiently worked out.

From the analysis of these comments it is clear that publication of your manuscript in our journal cannot be considered at this stage. On the other hand, given the potential interest of your study I would like to give you the opportunity to address the reviewers concerns and would be willing to consider a revised manuscript with the understanding that the referee concerns must be fully addressed and that acceptance of the manuscript would entail a second round of review.

I should also remind you that it is EMBO reports policy to allow a single round of revision only and that, therefore, acceptance or rejection of the manuscript will depend on the completeness of your responses included in the next, final version of the manuscript. I realize that addressing the referees comments in full - in particular the clarification of EGFR signaling, the link to autophagy and the physiological role of endophagy - would involve a lot of additional experimental work and I am uncertain whether you will be able to return a revised manuscript within our 3 months deadline. I could potentially further extend this period up to 5 months [should you feel time would be the only limitation to a successful revision of the paper] but would also understand your decision if you chose to rather seek rapid publication elsewhere at this stage.

Should you decide to embark on such a revision, revised manuscripts should be submitted within

three months of a request for revision; they will otherwise be treated as new submissions. Please contact us if a 3-months time frame is not sufficient for the revisions so that we can discuss the revisions further.

Supplementary/additional data: The Expanded View format, which will be displayed in the main HTML of the paper in a collapsible format, has replaced the Supplementary information. You can submit up to 5 images as Expanded View. Please follow the nomenclature Figure EV1, Figure EV2 etc. The figure legend for these should be included in the main manuscript document file in a section called Expanded View Figure Legends after the main Figure Legends section. Additional Supplementary material should be supplied as a single pdf labeled Appendix. The Appendix includes a table of content on the first page with page numbers, all figures and their legends. Please follow the nomenclature Appendix Figure Sx throughout the text and also label the figures according to this nomenclature. For more details please refer to our guide to authors.

Regarding data quantification, please ensure to specify the name of the statistical test used to generate error bars and P values, the number (n) of independent experiments underlying each data point (not replicate measures of one sample), and the test used to calculate p-values in each figure legend. Discussion of statistical methodology can be reported in the materials and methods section, but figure legends should contain a basic description of n, P and the test applied. Please also include scale bars in all microscopy images.

We now strongly encourage the publication of original source data with the aim of making primary data more accessible and transparent to the reader. The source data will be published in a separate source data file online along with the accepted manuscript and will be linked to the relevant figure. If you would like to use this opportunity, please submit the source data (for example scans of entire gels or blots, data points of graphs in an excel sheet, additional images, etc.) of your key experiments together with the revised manuscript. Please include size markers for scans of entire gels, label the scans with figure and panel number, and send one PDF file per figure.

- a complete author checklist, which you can download from our author guidelines (<http://embor.embopress.org/authorguide#revision>). Please insert page numbers in the checklist to indicate where the requested information can be found.
 - a letter detailing your responses to the referee comments in Word format (.doc)
 - a Microsoft Word file (.doc) of the revised manuscript text
 - editable TIFF or EPS-formatted figure files in high resolution
- (In order to avoid delays later in the publication process please check our figure guidelines before preparing the figures for your manuscript:
http://www.embopress.org/sites/default/files/EMBOPress_Figure_Guidelines_061115.pdf)
- a separate PDF file of any Supplementary information (in its final format)
 - all corresponding authors are required to provide an ORCID ID for their name. Please find instructions on how to link your ORCID ID to your account in our manuscript tracking system in our Author guidelines (<http://embor.embopress.org/authorguide>).

As part of the EMBO publication's Transparent Editorial Process, EMBO reports publishes online a Review Process File to accompany accepted manuscripts. This File will be published in conjunction with your paper and will include the referee reports, your point-by-point response and all pertinent correspondence relating to the manuscript.

I look forward to seeing a revised version of your manuscript when it is ready. Please let me know if you have questions or comments regarding the revision.

REFEREE REPORTS

Referee #1:

Fraser and co-workers report a novel role for the autophagy machinery in endosomal trafficking and growth factor receptor signaling. The authors observed an increase in PI(3)P positive early endosomes in glial cells lacking ATG7 or ATG16L, and EGFRs accumulated in these endosomes. In ATG7 KO cells, early endosomes were found to be positive for Galectin8, a marker for endosomal damage and recruiter of autophagy adaptors, and showed increased levels of ATG16L and WIPI2. Inhibition of TBK1, a kinase which phosphorylates autophagy receptors, or knockdown of Galectin8, prevented such recruitment. The authors propose that autophagy inhibition can reduce EGF-mediated signalling by prolonging the residence of EGFR on PI(3)P positive early endosomes, and that EGFR signaling is inhibited by increased EGF recycling to the plasma membrane. Overall this is a well-performed study that has provided several interesting results and conclusions. Experiments are accompanied by adequate quantifications, and the results look convincing. However, certain issues remain to be clarified.

Major points:

1. The authors show an increase in PI(3)P positive puncta in ATG7 and ATG16L defective cells, but it is not clear whether this reflects a redistribution of PI(3)P into smaller and more numerous vesicles, or a decrease in cellular PI(3)P. The authors should clarify this.
2. A puzzling finding was the observed decrease in endosomal EGF levels in ATG7 depleted cells, which was accompanied by decreased EGFR phosphorylation whereas total EGFR levels were unaffected and endosomal EGFR levels increased. The authors explain the decreased EGF labeling and EGFR signaling with decreased EGF recycling via the "slow" Rab11-dependent recycling route (whereas "fast" recycling mediated by Rab4 is unaffected). However, it is difficult to reconcile decreased recycling of EGF with decreased intracellular levels of EGF, and the authors need to provide an explanation for this.
3. The authors speculate that autophagy of damaged endosomes mediates a homeostatic regulation that results in reduced EGFR recycling to the plasma membrane. However, they provide no mechanistic explanation for the link between autophagy activation and receptor trafficking. In order to provide support for their model, they need to investigate whether the effects of autophagy inhibition on EGFR recycling and signaling can be exacerbated by agents that damage early endosomes.

Minor points:

1. It is interesting that Gal8, but not Gal3 or Gal9, was recruited to early endosomes in the absence of ATG7. Since all these galectins sense endosomal damage, the authors might want to speculate why only Gal8 is recruited in this case. Does it tell us something about the nature of the damage?
2. The recruitment of LAMP2 positive endosomes to damaged early endosomes by a mechanism which requires the autophagy machinery has led the authors to propose an endosome-phagy mechanism triggered by endosomal damage. The endosome-phagy hypothesis is a very interesting aspect of this work, which has been somewhat underplayed in the abstract and discussion. This is presumably because the authors plan to investigate this mechanism more in-depth. If so, electron microscopy would be essential in order to examine whether damaged endosomes become engulfed by double membranes.

Referee #2:

Fraser et al. report that deletion of ATG7 and some other ATGs causes accumulation of damaged early endosomes and defective EGFR recycling. They propose that delivery of damaged endosomes by TBK1- and Gal8-mediated recognition to autophagosomes for clearance (called "endosome-

phagy" in Discussion) is important for endosomal homeostasis. However, the reported observations do not tell a cohesive story. In particular, they do not convincingly show how much "endosome-phagy" contributes to the observed phenotype (i.e., accumulation of damaged endosomes and impaired EGFR recycling) and how it is different from the role of autophagy in lysosome activation. There are also other concerns as described below.

Major comments:

1. According to the authors' model, endosomes can become damaged spontaneously, even more so by monensin treatment etc, and subsequently cleared by autophagy. However, the relationship between this observation and the defect in EGFR recycling observed in ATG7 KO cells is unclear. Even if damaged endosomes are not degraded by autophagy, the remaining endosomes should be functional. It is possible that ATG7 has an independent function in the recycling pathway, but the authors do not properly distinguish these two functions. This point is also unclear in the model shown in Fig. 7G.
2. It is important to estimate the percentage of early endosomes that are turned over by autophagy under normal conditions (Fig. 7G). In Fig. 2, the number of endosomes increases upon deletion of ATG genes, indicating that a significant proportion of endosomes are constitutively disrupted and degraded by autophagy. However, this is not reflected in other images. For example, in Fig. 4C, there seems to be fewer EEA1 structures in ATG7 KO cells, particularly in the presence of monensin. The authors should also quantify the number of early endosomes in Gal8 knockout cells and TBK1-inhibited cells. The significance of this paper would be limited if the rate of the autophagy-dependent early endosome turnover turned out to be low.
3. Given that autophagic flux is important for lysosomal function (Scott et al. (2004) *Dev Cell* 7:167-178, Zhou et al. (2013) *Cell Res* 23:508-523), it is also critical to distinguish the role of "endosome-phagy" from that of autophagic flux in lysosome activation.
4. Any direct evidence of endosome autophagy (endosome-phagy) is not shown. Some typical electron microscopy images should be presented at least. It may be difficult but the conclusion would be considerably strengthened by electron microscopy images showing damaged endosomes in ATG knockout cells.
5. This reviewer has several concerns with the data on EGFR recycling. In ATG7 KO cells, EGFR internalization (Fig. 6C) and its total protein level are not altered (Fig. 7A), but its recycling to the plasma membrane is significantly impaired (Fig. 6D). These data suggest that EGFR should be trapped in some endosomal compartments. However, in ATG7 KO cells, its colocalization with Rab4 (fast recycling pathway) is not altered (Fig. S4H) and that with Rab11 (slow recycling pathway) is decreased (Fig. 6). Where is EGFR trapped? Also, while interesting, why recycling of the Tfn receptor is not affected should be explained.
6. This study uses several CRISPR-based knockout cells, but some of them are not cloned. The authors should show that the target genes (e.g., Gal8 and Atg16L1) are indeed deleted by immunoblotting (as shown for ATG7 in Fig. S1). Were these "bulk" knockout cells effectively selected using antibiotics or by flow cytometry?
7. The effect of ATG deletion should be confirmed using "rescued" cells. At least in key experiments (e.g., PI3P labeling, Gal8 staining, and EGFR signaling), the authors should use ATG knockout glial cells rescued by re-expression of the target genes.
8. The physiological role of this pathway is not convincing. Why is it cell-type specific?
9. The authors rule out the possibility of LAP by observing the normal recruitment of the ATG16L1K490A mutant. This is not valid. Although ATG16L1K490A cannot complement the LAP in ATG16L1 KO cells, it should be recruited to LAPsomes through binding to endogenous ATG16L1 when wild-type cells are used.

Minor comments:

1. The fluorescence images are generally too small to evaluate.
2. Why are Gal3 and Gal9 not involved? Are they recruited to monensin-treated endosomes?
3. The data in Fig. 3C should be quantified.

Referee #3:

The paper by Fraser et al entitled "Targeting of Early Endosomes by Autophagy Facilitates EGFR Recycling and Signalling" reports the key finding that (macro)autophagy is required for proper endocytic trafficking of the EGF receptor (EGFR) in a transformed glial cell model with Ink4a/Arf deletion and TP53 and Nf-1 knockdown. The authors present data suggesting that autophagy is required to degrade damaged early endosomes to ensure proper trafficking and signaling of the EGFR. In serum starved ATG7 KO cells stimulated with 20 ng/ml EGF the EGFR is trapped or halted in EEA1- and Rab5-positive early endosomes relatively to the situation in WT cells. Initial activation and endocytosis of the EGFR was not affected by autophagy loss. However, cells with KO of ATG7, ATG16L1, or ATG3, or Gal8, displayed reduced EGF-induced phosphorylation of EGFR, AKT, and ERK showing that signaling was compromised in autophagy deficient cells. These defects are caused because autophagy is required for endosomal quality control to ensure efficient endocytic recycling of EGFR to the plasma membrane via Rab11 recycling endosomes.

The paper is concise, very well written and the data nicely presented. The novelty aspect is also there since autophagy-mediated degradation of damaged early endosomes has not to my knowledge been reported or highlighted before although lysophagy and degradation of late endosomes have been extensively reported. EGFR-mediated signaling is of crucial importance and often dysregulated in cancer making the finding of autophagy regulating the recycling of EGFR of significance also beyond the fields of autophagy and endocytosis.

The data presented are for the most part convincing with relevant controls. However, some of the observations suggest a more cell-type specific effect on downstream signaling so it is not clear how general the effect on cell signaling downstream EGFR is.

1. Top of page 4: The authors have mixed complex 1 and 2 (or I and II) as the "autophagic" complex is complex I and the "endocytic complex is complex II.
2. In the Introduction when discussing "The endocytic pathway can also contribute to autophagosome biogenesis..." the authors could also include the finding that the ESCRT-III component CHMP2A has been implicated in phagophore closure during autophagosome biogenesis with ref to Takahashi et al. 2018 (Nat. Commun. PMID: 30030437).
3. There is a bit confusing mix of stimulation with EGF using 2 ng/ml and 20 ng/ml, sometimes even in the same figure panels. The authors could explain that stimulation with 2 ng/ml of EGF leads to receptor recycling whereas stimulation with 20 ng/ml leads to degradation of the EGFR in the lysosome and why they use the different concentrations for the different experiments.
4. The SIM experiment in in Fig. 4I is of purely qualitative nature. It would be good if the authors could provide some data on how statistically significant the result shown is.
5. In Fig 4, I miss triple staining experiments with EGFR, LC3 and EEA1 in WT cells stimulated or not with EGF and treated with (and) without monensin.
6. The signaling phenotype of pERK and pAkt activation downstream of EGFR was not the same in MEFs and in the glial cell line. It would be interesting if the authors explored this a bit more in a few more EGF-responsive cell lines perhaps using SAR405 and or ULK1 inhibitor . The cell death quantifications suggest a 2-fold increase in cell death which does not seem to be very dramatic. It would be more informative if the authors could relate their data to % dead cells in the populations for the reader to understand the magnitude of the effects observed.

1st Revision - authors' response

20 May 2019

Referee #1:

Fraser and co-workers report a novel role for the autophagy machinery in endosomal trafficking and

growth factor receptor signaling. The authors observed an increase in PI(3)P positive early endosomes in glial cells lacking ATG7 or ATG16L, and EGFRs accumulated in these endosomes. In ATG7 KO cells, early endosomes were found to be positive for Galectin8, a marker for endosomal damage and recruiter of autophagy adaptors, and showed increased levels of ATG16L and WIPI2. Inhibition of TBK1, a kinase which phosphorylates autophagy receptors, or knockdown of Galectin8, prevented such recruitment. The authors propose that autophagy inhibition can reduce EGF-mediated signalling by prolonging the residence of EGFR on PI(3)P positive early endosomes, and that EGFR signaling is inhibited by increased EGF recycling to the plasma membrane.

Overall this is a well-performed study that has provided several interesting results and conclusions. Experiments are accompanied by adequate quantifications, and the results look convincing. However, certain issues remain to be clarified.

Major points:

1. The authors show an increase in PI(3)P positive puncta in ATG7 and ATG16L defective cells, but it is not clear whether this reflects a redistribution of PI(3)P into smaller and more numerous vesicles, or a decrease in cellular PI(3)P. The authors should clarify this.

We thank the referee for this comment. We have clarified in the text that Figures 2A-B reflect measurements of total PI(3)P signal instead of puncta numbers thereby showing that cells lacking autophagy proteins exhibit higher cellular PI(3)P levels. The corresponding text was edited on pg. 5. We have also included a newly added Figure EV2B that shows PI(3)P staining at a lower magnification to further support the total increase in PI(3)P.

2. A puzzling finding was the observed decrease in endosomal EGF levels in ATG7 depleted cells, which was accompanied by decreased EGFR phosphorylation whereas total EGFR levels were unaffected and endosomal EGFR levels increased. The authors explain the decreased EGF labeling and EGFR signaling with decreased EGF recycling via the "slow" Rab11-dependent recycling route (whereas "fast" recycling mediated by Rab4 is unaffected). However, it is difficult to reconcile decreased recycling of EGF with decreased intracellular levels of EGF, and the authors need to provide an explanation for this.

We would like to clarify that in Figure 6D we have measured EGFR recycling and show that the receptor availability at the plasma membrane is reduced in autophagy-deficient cells. This suggests that the Rab11+ endosomes significantly contribute to EGFR recycling in our cells. The reduction in plasma membrane-localised EGFR can lead to diminished EGF uptake, and subsequently reduced receptor activation and downstream signalling in autophagy-deficient cells. These data are consistent with previous findings demonstrating that EGFR-plasma membrane localisation is important for EGF uptake and signalling (Eden et al., 2012; Sigismund et al., 2008). However, as EGFR degradation is not disrupted in the autophagy-deficient cells, we predict that intracellular EGF does not accumulate to levels that circumvents the decrease in EGF endocytosis despite the increased residence of EGFR at early endosomes. We have clarified this on pg. 8 of the revised manuscript.

3. The authors speculate that autophagy of damaged endosomes mediates a homeostatic regulation that results in reduced EGFR recycling to the plasma membrane. However, they provide no mechanistic explanation for the link between autophagy activation and receptor trafficking. In order to provide support for their model, they need to investigate whether the effects of autophagy inhibition on EGFR recycling and signaling can be exacerbated by agents that damage early endosomes.

In order to address the referee's comment, we have measured EGFR recycling and signalling in cells expressing or lacking ATG16L1 in the presence or absence of the endosome damaging agent, monensin. The data in Figure 6D show that EGFR recycling (calculated from endocytosed pool) is inhibited during monensin treatment to a similar level as in cells lacking autophagy or expressing EGFRvIII mutant. We did not observe a further exacerbation of EGFR recycling defect in the ATG16L1 knockout cells treated with monensin. This could be because early endosome damage induced by autophagy inhibition or monensin treatment may affect only a specific pool of early endosomes susceptible to damage. The remaining early endosomes that escape perturbation induced by chemical inhibitors or autophagy inhibition are still capable of driving lower rates of EGFR recycling. It is important to note, however, that there are currently no chemical molecules that can

specifically perturb early endosome function without interfering with other stages of vesicular trafficking. We observed that monensin treatment disrupted EGFR endocytosis (Figure 6C) which correlated with higher EGF-induced Akt signalling (as a result of stabilised EGFR at the plasma membrane, Rebuttal Figure 1 below). Enhanced Akt signalling was also observed in the absence of EGF stimulation further highlighting broad effects of this molecule on cellular signalling. Future development of chemical molecules that can specifically target early endosomes will be an interesting direction and may have important therapeutic potentials. The text was modified on pg. 8 to further discuss these findings.

Rebuttal Figure 1: Effects of monensin treatment on EGF-mediated signalling. Cells treated in the presence or absence of monensin were stimulated with EGF (for the indicated time points). Downstream signalling was then assessed by western blotting. As can be seen, monensin treatment leads to higher Akt phosphorylation suggestive of stabilised EGFR at the plasma membrane.

Minor points:

1. It is interesting that Gal8, but not Gal3 or Gal9, was recruited to early endosomes in the absence of ATG7. Since all these galectins sense endosomal damage, the authors might want to speculate why only Gal8 is recruited in this case. Does it tell us something about the nature of the damage? **The specific recruitment of Gal8 to early endosomes is indeed an intriguing question. We have expanded our discussion on pg. 11 to speculate the molecular relevance of this finding.**

2. The recruitment of LAMP2 positive endosomes to damaged early endosomes by a mechanism which requires the autophagy machinery has led the authors to propose an endosome-phagy mechanism triggered by endosomal damage. The endosome-phagy hypothesis is a very interesting aspect of this work, which has been somewhat underplayed in the abstract and discussion. This is presumably because the authors plan to investigate this mechanism more in-depth. If so, electron microscopy would be essential in order to examine whether damaged endosomes become engulfed by double membranes.

We thank the referee for this comment and agree that exploring endosome targeting by autophagosomes using EM would represent a convincing evidence. We have attempted to do this but were unable to obtain such images for several reasons. Firstly, given the unknown morphology of damaged endosomes, we were unable to conclude by conventional EM that they were indeed engulfed by autophagosomes in the absence of any label. In addition, we attempted to use an Au-EGF to mark early endosomes but were not successful in detecting any intracellular gold label by EM (likely due to technical reasons). Finally, in order to perform immuno-EM, we attempted to express GFP-tagged EEA1 to mark early endosomes, but observed that this construct led to the formation of large, morphologically abnormal vesicles when assessed under the fluorescent microscope (Rebuttal Figure 2 below). Because of these

limitations, and despite efforts from an EM expert (Chieko Kishi-Itakura, newly added co-author), we regret that we are unable to provide the requested EM images within the time frame limitation of this manuscript revision. We believe that our fluorescence colocalisation experiments and genetic deletion studies provide strong evidences for the targeting of endosomes by the autophagy machinery as presented in this study.

Rebuttal Figure 2: Localisation of GFP-EEA1. Glial cells expressing GFP-EEA1 were analysed by fluorescence microscopy. As can be seen, exogenous GFP-EEA1 formed large vacuolated structures that do not resemble early endosome morphology.

Referee #2:

Fraser et al. report that deletion of ATG7 and some other ATGs causes accumulation of damaged early endosomes and defective EGFR recycling. They propose that delivery of damaged endosomes by TBK1- and Gal8-mediated recognition to autophagosomes for clearance (called "endosome-phagy" in Discussion) is important for endosomal homeostasis. However, the reported observations do not tell a cohesive story. In particular, they do not convincingly show how much "endosome-phagy" contributes to the observed phenotype (i.e., accumulation of damaged endosomes and impaired EGFR recycling) and how it is different from the role of autophagy in lysosome activation. There are also other concerns as described below.

Major comments:

1. According to the authors' model, endosomes can become damaged spontaneously, even more so by monensin treatment etc, and subsequently cleared by autophagy. However, the relationship between this observation and the defect in EGFR recycling observed in ATG7 KO cells is unclear. Even if damaged endosomes are not degraded by autophagy, the remaining endosomes should be functional. It is possible that ATG7 has an independent function in the recycling pathway, but the authors do not properly distinguish these two functions. This point is also unclear in the model shown in Fig. 7G.

We thank the referee for this comment. Our data show that damaged endosomes can trap a subpopulation of EGFR destined to be recycled to the plasma membrane. This appears to be sufficient to reduce receptor recycling, although as indicated by the referee, remaining intact endosomes are capable of driving a lower level of recycling (potentially through Rab4+ endosomes). We have addressed the relationship between damaged endosomes and reduced EGFR recycling (as also requested by referee #1 point 3) by providing evidence that receptor recycling is disrupted upon treatment with monensin (Figure 6D). On the other hand, we have addressed whether the effects of reduced EGFR recycling are due to autophagy-independent activities of ATG7 by showing that the inhibition of autophagy by ATG16L1 deletion similarly leads to a decrease in receptor recycling (Figure 6D). These data provide further evidence that the intact autophagy machinery is required for efficient recycling of EGFR.

2. It is important to estimate the percentage of early endosomes that are turned over by autophagy under normal conditions (Fig. 7G). In Fig. 2, the number of endosomes increases upon deletion of ATG genes, indicating that a significant proportion of endosomes are constitutively disrupted and

degraded by autophagy. However, this is not reflected in other images. For example, in Fig. 4C, there seems to be fewer EEA1 structures in ATG7 KO cells, particularly in the presence of monensin. The authors should also quantify the number of early endosomes in Gal8 knockout cells and TBK1-inhibited cells. The significance of this paper would be limited if the rate of the autophagy-dependent early endosome turnover turned out to be low.

We would like to clarify that the shown images do not directly represent the number of early endosomes which is variable between cells of the same population. We have quantified the number of early endosomes and observed that a potential increase in EEA1-positive endosomes in autophagy-deficient cells was not significant when compared to control cells (newly added Figure EV3C). The lack of significant increase can be due to the high variability in endosome numbers between cells of the same population or due to the existence of compensatory mechanisms that regulate endosome biogenesis and maturation (beyond the scope of this manuscript).

To address the referee's comment and to examine the rate of early endosome targeting under normal conditions, we treated cells acutely with Bafilomycin A1 to inhibit lysosomal degradation and assessed the incidence of LC3-positive early endosomes. The newly generated data (Figures 4A and 4B) show that a significant percentage of early endosomes stain positive for LC3 after 1 hour treatment with Bafilomycin A1 suggesting that continual targeting of early endosomes does occur under basal conditions.

3. Given that autophagic flux is important for lysosomal function (Scott et al. (2004) *Dev Cell* 7:167-178, Zhou et al. (2013) *Cell Res* 23:508-523), it is also critical to distinguish the role of "endosome-phagy" from that of autophagic flux in lysosome activation.

In order to measure lysosomal activity in cells used in our study, we utilised LysoSensor probe to measure lysosomal acidification induced by amino acid starvation (as performed by Zhou et al., 2013). The newly added Figure EV4C shows that lysosomal activation remains intact in the absence of autophagy in our system. This is in agreement with our data showing that the lysosomal degradation of EGFR is not affected in autophagy-deficient cells (Figure EV4D).

4. Any direct evidence of endosome autophagy (endosome-phagy) is not shown. Some typical electron microscopy images should be presented at least. It may be difficult but the conclusion would be considerably strengthened by electron microscopy images showing damaged endosomes in ATG7 knockout cells.

We thank the referee for this comment. This has been addressed above as suggested by referee #1, minor point 2.

5. This reviewer has several concerns with the data on EGFR recycling. In ATG7 KO cells, EGFR internalization (Fig. 6C) and its total protein level are not altered (Fig. 7A), but its recycling to the plasma membrane is significantly impaired (Fig. 6D). These data suggest that EGFR should be trapped in some endosomal compartments. However, in ATG7 KO cells, its colocalization with Rab4 (fast recycling pathway) is not altered (Fig. S4H) and that with Rab11 (slow recycling pathway) is decreased (Fig. 6). Where is EGFR trapped? Also, while interesting, why recycling of the Tfn receptor is not affected should be explained.

We thank the referee for raising this important point which requires further clarification in our manuscript. Our data demonstrate that EGFR is trapped in a subset of early endosomes (Figures 1F and 2D) which are unable to progress into Rab11+ recycling endosomes. We have expanded our interpretation of the differential reliance of Tfn and EGFR trafficking on autophagy. The text has been modified on pg. 11 to further clarify these points.

6. This study uses several CRISPR-based knockout cells, but some of them are not cloned. The authors should show that the target genes (e.g., Gal8 and Atg16L1) are indeed deleted by immunoblotting (as shown for ATG7 in Fig. S1). Were these "bulk" knockout cells effectively selected using antibiotics or by flow cytometry?

We have included confirmation of ATG16L1 and Gal8 loss of expression in the newly added Figures EV2A and 5E, as suggested by the referee. These knockouts were also confirmed in Figure 7. It is important to note that a robust knockout of the target gene expression in glial cells was achieved by expressing Cas9 and gRNA sequences using viral systems (clarified on pg. 13 under material and methods section). In all cases, bulk cells were analysed instead of single

cloned lines. We have also highlighted in the material and methods section that the knockout lines were assessed early on after their generation and were not passaged for prolonged periods.

7. The effect of ATG deletion should be confirmed using "rescued" cells. At least in key experiments (e.g., PI3P labeling, Gal8 staining, and EGFR signaling), the authors should use ATG knockout glial cells rescued by re-expression of the target genes.

In our study, we used two different guide RNA sequences targeting each of the multiple endosome-phagy players (including ATG7, ATG3, ATG16L1, ATG13, and Gal8) as means to inhibit autophagy in glial cells. We believe that this provides sufficient control for any potential off-target and autophagy-independent effects. Due to the viral method used to generate these lines and the guide RNA designed to target the coding sequences, we regret that performing reconstitution experiments is not feasible in our system (due to the targeting of exogenously expressed sequences by Cas9/gRNA stable expression) and in the time allocation of this manuscript revision.

8. The physiological role of this pathway is not convincing. Why is it cell-type specific?

We thank the referee for this comment, which we have addressed below as also suggested by referee #3, point 6.

9. The authors rule out the possibility of LAP by observing the normal recruitment of the ATG16L1K490A mutant. This is not valid. Although ATG16L1K490A cannot complement the LAP in ATG16L1 KO cells, it should be recruited to LAPsomes through binding to endogenous ATG16L1 when wild-type cells are used.

In order to address this referee's comment, we expressed ATG16L1 K490A mutant in ATG16L1 knockout MEFs and show that it can also be targeted to early endosomes during monensin treatment (newly added Figures EV3G and EV3H). It is also important to note that previous studies have shown that the overexpression of ATG16L1 can suppress endogenous ATG16L1 levels (Fujita et al., 2013; Fujita et al., 2009). The requirement for the canonical autophagy machinery was also confirmed by the finding that treatment with Bafilomycin A1 (previously shown to inhibit LAP-like processes) does not influence the recruitment of GFP-LC3 to early endosomes (newly added Figures 4A and 4B). Furthermore, we show that knockdown of ATG13 (dispensable for LAP) similarly disrupts EGF-mediated signalling (newly added Figure EV5B). Altogether, our data support that the canonical autophagy machinery can target early endosomes and are required for optimal signalling during EGF treatment.

Minor comments:

1. The fluorescence images are generally too small to evaluate.

As suggested by the referee, we have increased the sizes of the fluorescence images throughout the manuscript.

2. Why are Gal3 and Gal9 not involved? Are they recruited to monensin-treated endosomes?

The specific recruitment of Gal8 to early endosomes is indeed an intriguing question. As also suggested by Referee #1 (minor point #1), we have expanded our discussion on pg. 11 to speculate the molecular relevance of this finding.

3. The data in Fig. 3C should be quantified.

We have included the quantification of Figure 3C in the newly added Figure 3D as suggested by the referee.

Referee #3:

The paper by Fraser et al entitled "Targeting of Early Endosomes by Autophagy Facilitates EGFR Recycling and Signalling" reports the key finding that (macro)autophagy is required for proper endocytic trafficking of the EGF receptor (EGFR) in a transformed glial cell model with Ink4a/Arf deletion and TP53 and Nf-1 knockdown. The authors present data suggesting that autophagy is required to degrade damaged early endosomes to ensure proper trafficking and signaling of the EGFR. In serum starved ATG7 KO cells stimulated with 20 ng/ml EGF the EGFR is trapped or halted in EEA1- and Rab5-positive early endosomes relatively to the situation in WT cells. Initial

activation and endocytosis of the EGFR was not affected by autophagy loss. However, cells with KO of ATG7, ATG16L1, or ATG3, or Gal8, displayed reduced EGF-induced phosphorylation of EGFR, AKT, and ERK showing that signaling was compromised in autophagy deficient cells. These defects are caused because autophagy is required for endosomal quality control to ensure efficient endocytic recycling of EGFR to the plasma membrane via Rab11 recycling endosomes.

The paper is concise, very well written and the data nicely presented. The novelty aspect is also there since autophagy-mediated degradation of damaged early endosomes has not to my knowledge been reported or highlighted before although lysophagy and degradation of late endosomes have been extensively reported. EGFR-mediated signaling is of crucial importance and often dysregulated in cancer making the finding of autophagy regulating the recycling of EGFR of significance also beyond the fields of autophagy and endocytosis.

The data presented are for the most part convincing with relevant controls. However, some of the observations suggest a more cell-type specific effect on downstream signaling so it is not clear how general the effect on cell signaling downstream EGFR is.

1. Top of page 4: The authors have mixed complex 1 and 2 (or I and II) as the "autophagic" complex is complex I and the "endocytic complex is complex II.

We thank the referee for pointing out this error. We have corrected this as specified on pg. 3.

2. In the Introduction when discussing "The endocytic pathway can also contribute to autophagosome biogenesis..." the authors could also include the finding that the ESCRT-III component CHMP2A has been implicated in phagophore closure during autophagosome biogenesis with ref to Takahashi et al. 2018 (Nat. Commun. PMID: 30030437).

We thank the referee for this suggestion and have referenced the suggested work on pg. 4 of the manuscript.

3. There is a bit confusing mix of stimulation with EGF using 2 ng/ml and 20 ng/ml, sometimes even in the same figure panels. The authors could explain that stimulation with 2 ng/ml of EGF leads to receptor recycling whereas stimulation with 20 ng/ml leads to degradation of the EGFR in the lysosome and why they use the different concentrations for the different experiments.

We have included a description of our choice of EGF concentrations on pg. 13 of the revised manuscript. Importantly, we clarified that the effects seen on targeting of EGFR+ endosomes upon autophagy inhibition are not dependent on EGF concentrations.

4. The SIM experiment in in Fig. 4I is of purely qualitative nature. It would be good if the authors could provide some data on how statistically significant the result shown is.

As per the referee's suggestion, we have quantified the SIM data in the newly added Figure 4J. Due to the low throughput nature of SIM, we could only quantify a relatively low number of cells (specified in the figure legend).

5. In Fig 4, I miss triple staining experiments with EGFR, LC3 and EEA1 in WT cells stimulated or not with EGF and treated with (and) without monensin.

As suggested by the referee, we assessed by SIM structures positive for EEA1 and 555-EGF as well as ATG16L1 (as an autophagy marker) in cells treated with monensin and included the data in Figure 4I. Since in the absence of monensin treatment, punctate structures formed by autophagy players are very rare, we did not include these data.

6. The signaling phenotype of pERK and pAkt activation downstream of EGFR was not the same in MEFs and in the glial cell line. It would be interesting if the authors explored this a bit more in a few more EGF-responsive cell lines perhaps using SAR405 and or ULK1 inhibitor. The cell death quantifications suggest a 2-fold increase in cell death which does not seem to be very dramatic. It would be more informative if the authors could relate their data to % dead cells in the populations for the reader to understand the magnitude of the effects observed.

We agree with the referee that the cell type specific differences in response to EGF stimulation is intriguing. It is important to highlight that our data suggest that targeting of early endosomes by autophagy occurs in the two cell lines tested in this study and is thereby likely to occur in a wide panel of cell types. However, the downstream response to EGF stimulation depends on oncogenic mutations or EGFR levels. We further explored the effects of autophagy

inhibition in a panel of cell lines during EGF stimulation (newly added Figures 7G and EV5F-I). We chose to genetically inhibit autophagy instead of using chemical inhibitors to avoid any potential non-specific effects of these inhibitors. Interestingly, we observed a strong reduction in pEGFR (receptor activation) or total EGFR levels across the majority of cell lines tested indicating that autophagy is important for the proper trafficking and activation of EGFR. Similar to the differences in EGF stimulation response observed between MEFs and glial cells, we observed a variation in downstream signalling, potentially dependent on additional oncogenic mutations that promote growth factor signalling in these cells. We further expanded the text on pg. 10 to discuss this.

As also suggested by this referee, we performed the cell death assay during serum withdrawal and EGF treatment and quantified the data as percentage values (included as Figure 7H, whereas the previous data were moved to Figures EV5J and EV5K). The newly performed and analysed data further support the conclusion that autophagy is required to maintain EGF-mediated cell survival.

References

- Eden, E.R., Huang, F., Sorkin, A., and Futter, C.E. (2012). The role of EGF receptor ubiquitination in regulating its intracellular traffic. *Traffic* *13*, 329-337.
- Fujita, N., Morita, E., Itoh, T., Tanaka, A., Nakaoka, M., Osada, Y., Umemoto, T., Saitoh, T., Nakatogawa, H., Kobayashi, S., *et al.* (2013). Recruitment of the autophagic machinery to endosomes during infection is mediated by ubiquitin. *J Cell Biol* *203*, 115-128.
- Fujita, N., Saitoh, T., Kageyama, S., Akira, S., Noda, T., and Yoshimori, T. (2009). Differential involvement of Atg16L1 in Crohn disease and canonical autophagy: analysis of the organization of the Atg16L1 complex in fibroblasts. *J Biol Chem* *284*, 32602-32609.
- Sigismund, S., Argenzio, E., Tosoni, D., Cavallaro, E., Polo, S., and Di Fiore, P.P. (2008). Clathrin-mediated internalization is essential for sustained EGFR signaling but dispensable for degradation. *Dev Cell* *15*, 209-219.
- Zhou, J., Tan, S.H., Nicolas, V., Bauvy, C., Yang, N.D., Zhang, J., Xue, Y., Codogno, P., and Shen, H.M. (2013). Activation of lysosomal function in the course of autophagy via mTORC1 suppression and autophagosome-lysosome fusion. *Cell Res* *23*, 508-523.

2nd Editorial Decision

5 July 2019

Thank you for the submission of your revised manuscript to EMBO reports. Unfortunately, former referee #1 was not available anymore but we have now received the reports from referee #2 and #3, which are copied below.

As you will see, both referees are positive about the study and support publication after some clarification to figures and text. Referee 2 asks for a better representation of the EEA1 data and the quantification shown in Fig. 4C and EV3C. Moreover, in the absence of convincing evidence that damaged endosomes are engulfed by autophagosomes (EM data), the corresponding conclusions should be toned down and alternative scenarios discussed. Please address these and the other remaining referee concerns in a final revision.

From the editorial side, there are also a few things that we need before we can proceed with the official acceptance of your study:

- Chieko Kishi-Itakura is not listed in the Author Contributions
- Fig EV5F: the blot for ATG7 appears overcontrasted. Please provide a scan with less contrast modification.
- I inspected the figures and their legends for completeness and accuracy. Please see the attached Word file for suggestions regarding the legends and the Abstract (in track changes/comments).

REFEREE REPORTS

Referee #2:

While the authors did make a great effort to respond to the criticisms previously made by the reviewers by including additional experimental data, there are still several concerns.

#2

The authors now suggest that ~5% of EEA1-positive endosomes could be subjected to autophagy per hour under normal conditions (Fig. 4B). However, the number of EEA1-positive endosomes does not significantly increase in sgATG7 cells despite the tendency to (Fig. EV3C). This reviewer suggests that this tendency should be reflected in images in Figure 4C by replacing them with more representative ones (the current images contain fewer structures). It is also recommended to plot all the data points (not only showing mean values and error bars) with actual p values rather than "NS" in the statistical analysis (e.g., in Fig. EV3C). Overall, the statistical information is not sufficient in the current manuscript.

#4

It is a bit unfortunate that the authors fail to provide electron microscopic evidence of "endosome-phagy". This outcome is rather surprising, given that the rate of endosome-phagy could be as high as ~5%/h under normal conditions (Fig. 4B). Perhaps, this issue could be better addressed by CLEM. Without EM data, the authors should be more careful to conclude that damaged endosomes are indeed engulfed by autophagosomes and that the observed phenomenon is not due to recruitment of LC3 to the endosomal membrane itself or autophagosomes fusing with endosomes to form amphisomes. Given that convincing morphological evidence is not presented, the authors should discuss these other possibilities.

Referee #3:

The authors have carefully addressed and answered all the referee's comments. Several new experiments and analysis have been included in the revised version of the manuscript. EM imaging to visualize engulfment of damaged endosomes by autophagosomes was apparently extensively attempted, but without success. Overall, the concerns raised by the referee's have been adequately answered.

Minor detail: In Figure 2D the duration time for the upper panel (5 min) and the lower panel (15 min) is not indicated in the figure (as it is e.g. in Figure 1B and 1F).

2nd Revision - authors' response

19 July 2019

Referee #2:

While the authors did make a great effort to respond to the criticisms previously made by the reviewers by including additional experimental data, there are still several concerns.

#2. The authors now suggest that ~5% of EEA1-positive endosomes could be subjected to autophagy per hour under normal conditions (Fig. 4B). However, the number of EEA1-positive endosomes does not significantly increase in sgATG7 cells despite the tendency to (Fig. EV3C). This reviewer suggests that this tendency should be reflected in images in Figure 4C by replacing them with more representative ones (the current images contain fewer structures). It is also recommended to plot all the data points (not only showing mean values and error bars) with actual p values rather than "NS" in the statistical analysis (e.g., in Fig. EV3C). Overall, the statistical information is not sufficient in the current manuscript.

As suggested by the referee, we replaced the images in Fig 4C to reflect a potential increase in EEA1-positive endosomes when autophagy is inhibited. Furthermore, we replaced the quantification in Fig EV3C to plot the individual data points as well as specified the p values in the figure and legend.

#4. It is a bit unfortunate that the authors fail to provide electron microscopic evidence of "endosome-phagy". This outcome is rather surprising, given that the rate of endosome-phagy could be as high as ~5%/h under normal conditions (Fig. 4B). Perhaps, this issue could be better addressed by CLEM. Without EM data, the authors should be more careful to conclude that damaged endosomes are indeed engulfed by autophagosomes and that the observed phenomenon is not due to recruitment of LC3 to the endosomal membrane itself or autophagosomes fusing with endosomes to form amphisomes. Given that convincing morphological evidence is not presented, the authors should discuss these other possibilities.

As suggested by the referee, we included a discussion of other possible outcomes of the association between autophagy proteins and endosomes. These have been included in the discussion on pg.12 of the revised manuscript (text highlighted in bold blue font).

Referee #3:

The authors have carefully addressed and answered all the referee's comments. Several new experiments and analysis have been included in the revised version of the manuscript. EM imaging to visualize engulfment of damaged endosomes by autophagosomes was apparently extensively attempted, but without success. Overall, the concerns raised by the referee's have been adequately answered.

Minor detail: In Figure 2D the duration time for the upper panel (5 min) and the lower panel (15 min) is not indicated in the figure (as it is e.g. in Figure 1B and 1F).

We have modified Fig 2D to include the EGF treatment in the upper and lower panels as suggested by the referee.

Corresponding Author Name: Noor Gammoh

Manuscript Number: EMBOR-2019-47734V1